# Thermodynamic and Kinematic Drivers of Atmospheric Boundary Layer Stability in the Central Arctic during MOSAiC

Gina C. Jozef[1,2,3], John J. Cassano[1,2,3], Sandro Dahlke[4], Mckenzie Dice[1,2,3], Christopher J. Cox[5], Gijs de Boer[2,5,6]

[1]Dept. of Atmospheric and Oceanic Sciences, University of Colorado Boulder, Boulder, CO, USA
[2]Cooperative Institute for Research in Environmental Sciences, University of Colorado Boulder, Boulder, CO, USA
[3]National Snow and Ice Data Center, University of Colorado Boulder, Boulder, CO, USA
[4]Alfred Wegener Institute Helmholtz Centre for Polar and Marine Research, Potsdam, Germany
[5]NOAA Physical Sciences Laboratory, Boulder, CO, USA
[6]Integrated Remote and In Situ Sensing, University of Colorado Boulder, Boulder, CO, USA

*Correspondence to:* Gina Jozef (gina.jozef@colorado.edu)

**Abstract.** Observations collected during the Multidisciplinary drifting Observatory for the Study of Arctic Climate (MOSAiC) provide a detailed description of the impact of thermodynamic and kinematic forcings on atmospheric boundary layer (ABL) stability in the central Arctic. This study reveals that the Arctic ABL is stable and near-neutral with similar frequencies, and strong stability is the most persistent of all stability regimes. MOSAiC radiosonde observations, in conjunction with observations from additional measurement platforms including a 10 m meteorological tower, ceilometer, microwave radiometer, and radiation station, provide insight into the relationships between atmospheric stability and various atmospheric thermodynamic and kinematic forcings of ABL turbulence, and how these relationships differ by season. We found that stronger stability largely occurs in low wind (i.e., wind speeds are slow), low radiation (i.e., surface radiative fluxes are minimal) environments, a very shallow mixed ABL forms in low wind, high radiation environments, weak stability occurs in high wind, moderate radiation environments, and a near-neutral ABL forms in high wind, high radiation environments. Surface pressure (a proxy for synoptic staging) partially explains the observed wind speeds for different stability regimes. Cloud frequency and atmospheric moisture contribute to the observed surface radiation budget. Unique to summer, stronger stability may also form when moist air is advected from over the warmer open ocean to over the colder sea ice surface, which decouples the colder near-surface atmosphere from the advected layer, and is identifiable through observations of fog and atmospheric moisture.

## 1 Introduction

The structure of the atmospheric boundary layer (ABL), which is the turbulent lowest part of the atmosphere that is directly influenced by the earth's surface (Stull, 1988; Marsik et al., 1995), affects the transfer of energy, moisture, and momentum between the Earth's surface and the overlying atmosphere (Brooks et al., 2017). A lack of detailed understanding of ABL structure over Arctic sea ice results from a historical shortage of the necessary in situ measurements. This study utilizes newly available high temporal and vertical spatial resolution atmospheric observations from the Multidisciplinary drifting Observatory for the Study of Arctic Climate (MOSAiC; Shupe et al., 2020) to analyze the relationships between atmospheric stability and the key thermodynamic and kinematic processes

dominating the Arctic ABL, primarily radiation (influenced by cloud cover) and wind shear, and how these relationships differ by season.

In the central Arctic, turbulence and static stability in the ABL are typically either mechanically and/or radiatively driven. Mechanical processes impacting the Arctic ABL include the interaction between the atmosphere and surface roughness features such as ridges and ice edges (Andreas et al., 2010) or oceanic waves (Jenkins et al., 2012), or the presence of a low-level jet (Brooks et al., 2017; Banta et al., 2003) which enhances wind shear below the jet core. Measurements of near-surface wind speed can be used to infer mechanical production of turbulence (Banta, 2008). Radiatively influenced processes impacting the Arctic ABL include the generation of buoyant turbulence through surface energy fluxes emitted from open water regions such as leads (Lüpkes et al., 2008), cold air advection, especially over thin ice (Vihma et al., 2005), enhanced downwelling longwave radiation from low level clouds (Wang et al., 2001), or turbulent mixing within the clouds and below cloud base due to cloud-top radiative cooling (Tjernström et al., 2004; Chechin et al., 2023). Measurements of the surface radiation budget and cloud characteristics support an understanding of the possibility for radiatively generated turbulence in the ABL. Due to the relatively reflective surfaces found in the central Arctic, solar heating of the Earth's surface and the resulting formation of buoyant thermals, which is a dominant forcing of the ABL in most parts of the planet (Marsik et al., 1995), plays only a minor role in the Arctic.

Previous studies have shown that the Arctic ABL is typically either stable or near-neutral, and a convective ABL is rarely observed (Brooks et al., 2017; Tjernström and Graversen, 2009; Persson et al., 2002; Esau and Sorokina, 2010). A stable ABL is typically observed when winds are light and when there is negative net longwave radiation at the surface (i.e., in the absence of clouds, or if clouds are very high; Stull, 1988)), as turbulence is weak and intermittent (Banta, 2008); this is common in Arctic winter (Tjernström and Graversen, 2009). However, a stable ABL may also form in the presence of low clouds and resulting enhanced downwelling longwave radiation when warm air is advected over the colder ice surface, contributing to a persistent fog layer above the sea ice, and decoupling of a shallow stable ABL from the advected layer above (Tjernström, 2005); this is common in Arctic summer (Tjernström et al., 2019).

A weakly stable or near-neutral atmosphere is expected in the presence of faster near-surface winds and when enhanced downwelling longwave radiation caused by cloud cover (particularly low clouds containing liquid water) erodes the surface inversion through radiative mixing, which is sometimes enhanced by downward mixing from the cloud itself (forced by cloud-top radiative cooling) (Vihma et al., 2005). Such clouds have a warming influence on the surface for most of the year (Brooks et al., 2017; Shupe and Intrieri, 2004). Only for a brief period in summer do clouds have a net cooling effect on the surface, when their blocking of incoming solar radiation outweighs their longwave warming effect (Shupe and Intrieri, 2004).

The processes described above are part of the complex ABL dynamics, and together have important implications for sea ice thickness and extent. Thus, to properly represent the central Arctic in weather and climate models, the relationships between radiatively and mechanically driven turbulence and ABL stability, and the seasonal differences, must be documented. While previous work does reveal some important information about the Arctic ABL features

and processes, most in situ observations have either been brief, located near the coast, or have only included measurements of a subset of important atmospheric features. Particularly lacking have been observations of atmospheric properties during the winter, as few previous field campaigns have gathered wintertime Arctic observations (e.g., the Surface Heat Budget of the Arctic Ocean (SHEBA) project; Uttal et al., 2002). MOSAiC obtained the necessary data from the central Arctic ice pack, between September 2019 and October 2020, to analyze atmospheric thermodynamic and kinematic features related to the ABL above the sea ice pack, from deep in the pack ice to near the marginal ice zone.

The questions guiding this study are as follows: What are the stability regimes present and their relative frequencies, annually and seasonally? What are the important relationships between thermodynamic and kinematic features present in the lower atmosphere, and ABL stability? How do these relationships differ by season? We hypothesize that wind speed and the surface radiation budget (which is strongly influenced by cloud cover) differ depending on ABL stability, but the relationships differ by season.

To determine the range of ABL stability and identify important thermodynamic and kinematic features in the Arctic ABL, we primarily use profile data from radiosondes launched at least four times per day throughout the entire MOSAiC year. First, we group the radiosonde observations based on stability to determine the relative frequency of occurrence of the various stability regimes, and how the regimes transition between each other. Then, we analyze how these regimes relate to wind speed, surface radiation budget, and atmospheric moisture, measured from a meteorological tower, radiation station, ceilometer, and microwave radiometer, in the context of ABL stability. We also assess the seasonal shifts in these characteristics and provide explanations for the observed thermodynamic and kinematic features.

## 2 Methods

### 2.1 Observational data from MOSAiC

Data used in this study were collected during MOSAiC, a year-long icebreaker-based expedition lasting from September 2019 through October 2020, during which the Research Vessel *Polarstern* (Alfred-Wegener-Institut Helmholtz-Zentrum für Polar- und Meeresforschung, 2017) was frozen into the central Arctic Ocean sea ice pack, and was set to drift passively across the central Arctic for the entire year. However, between 17 May and 18 June, between 31 July and 21 August, and between 21 September and 1 October 2020, it was necessary (for logistical reasons) for the *Polarstern* to travel under its own power. During the MOSAiC year, many measurements were taken to observe the atmosphere (Shupe et al. 2022), sea ice (Nicolaus et al. 2022), and ocean (Rabe et al. 2022), with the result being the most comprehensive observations of the central Arctic climate system to date. These measurements span all seasons, as well as both far from and close to the sea ice edge, as the *Polarstern* essentially followed one ice floe for its annual life cycle (only relocating to a new ice floe for the final two months of the expedition).

For this study, we primarily use profile data from the balloon-borne Vaisala RS41 radiosondes, which were launched from the stern deck of the *Polarstern* (~12 m above sea level) at least four times per day (every 6 hours), typically at 05:00, 11:00, 17:00, and 23:00 UTC (Maturilli et al., 2021). We use the level 2 radiosonde product for this analysis, as the level 2 data are found to be more reliable in the lower troposphere than the level 3 data (see the abstracts for the level 2 (Maturilli et al., 2021) and level 3 (Maturilli et al., 2022) data for explanation of the difference between the two options). Figure 1 shows the location of each radiosonde launch throughout the MOSAiC year. From the radiosondes, we utilize measurements of temperature, pressure, relative humidity, and wind speed and direction, as well as derived measurements of virtual potential temperature ($\theta_v$) and mixing ratio. The radiosondes ascend at a rate of approximately 5 m s$^{-1}$, sampling with a frequency of 1 Hz, which results in measurements about every 5 m throughout the ascent.

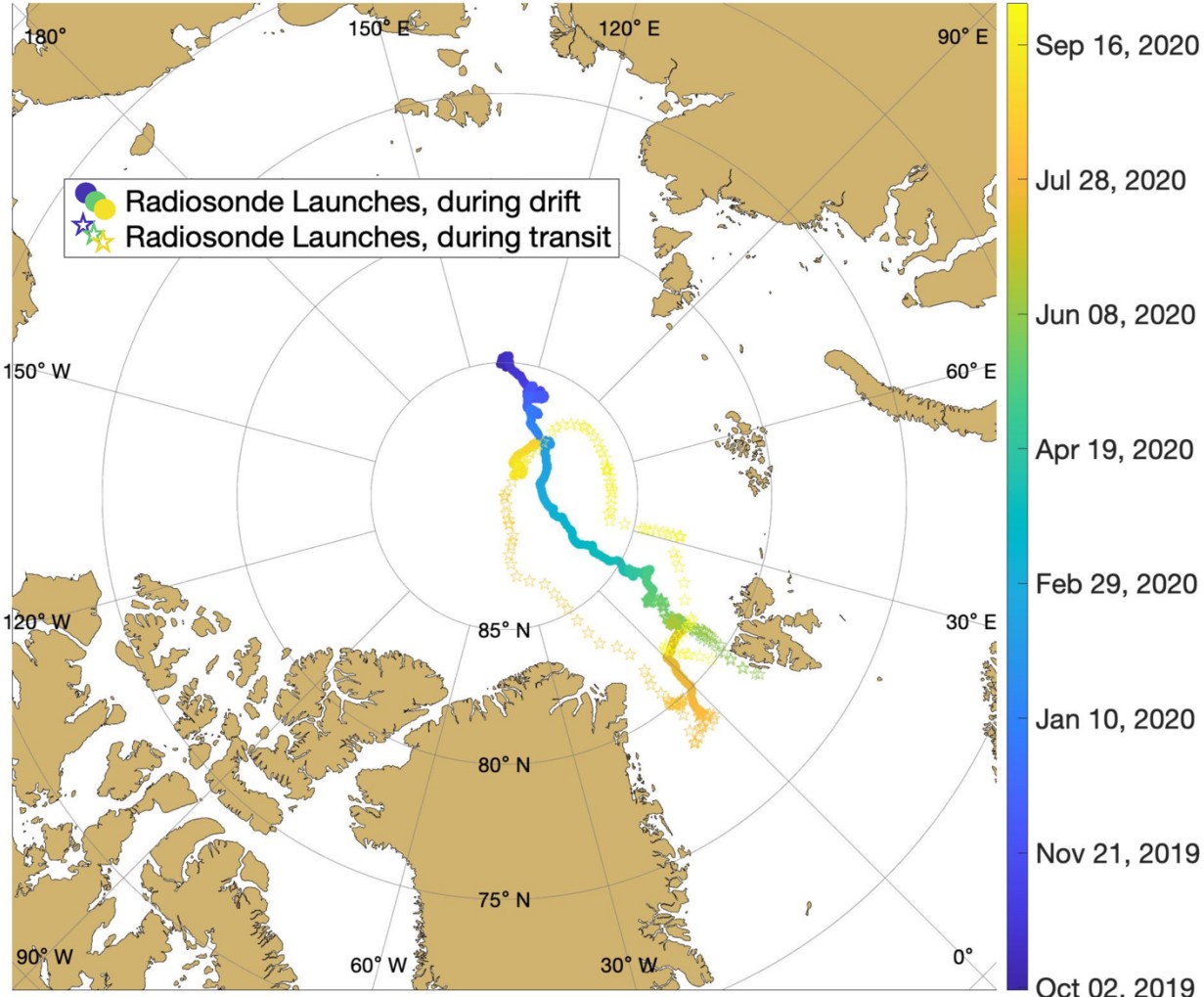

**Figure 1.** Map of the central Arctic showing the location of each radiosonde launch, color coded by date. Circular symbols indicate when the *Polarstern* was passively drifting, and star symbols indicate when the *Polarstern* was travelling under its own power.

In addition to the profile data provided by the radiosondes, we utilize observations from several surface-based platforms. Atmospheric observations of wind speed and pressure at 2 m above the surface come from a 10 m meteorological tower (hereafter "met tower"; Cox et al., 2023a) located on the sea ice near the *Polarstern* (Cox et al., 2023b), and provide information about near-surface mechanical mixing potential and synoptic setting at the time of each radiosonde launch. Pressure tendency corresponding to each radiosonde observation was calculated as the change in 2 m pressure over the three hours preceding the radiosonde launch. Several additional measurements come from instrumentation deployed as part of the Atmospheric Radiation Measurement (ARM) mobile facility (Shupe et al., 2021). Information on cloud cover comes from a Vaisala Ceilometer CL31 (ARM user facility, 2019a), which measures atmospheric backscatter and cloud base height (CBH), and allows us to determine the altitude and frequency of clouds at and before radiosonde launch. Additionally, precipitable water vapor (PWV) comes from the MWRRET Value-added Product (ARM user facility, 2019b) which derives PWV from ARM 2-channel microwave radiometer-measured brightness temperatures. PWV derivation and uncertainty are discussed in Turner et al. (2007) and Cadeddu et al. (2013) respectively. Both the ceilometer and microwave radiometer were located on the P-deck of the *Polarstern* (depicted in Fig. 3 of Shupe et al. 2022), which is approximately 20 m above sea level, and could occasionally be above a layer of fog. Thus, to identify periods of fog, we use meteorological observations manually reported by a designated weather observer onboard the *Polarstern*, which comply with the standards of the World Meteorological Organization and the German Weather Service (Schmithüsen and Raeke, 2021a, b, and c).

Lastly, measurements of upwelling and downwelling longwave and shortwave radiation come from an Eppley Precision Infrared Radiometer and Eppley Standard Precision Pyranometer deployed on the sea ice near the met tower (Cox et al., 2023b). Table 1 lists the instrument name and uncertainty for each of the observational variables used in this study.

**Table 1:** Instrument name and uncertainty for each variable used in this study.

| Platform | Variable | Instrumentation | Uncertainty |
|---|---|---|---|
| **Radiosonde** | Pressure | Vaisala RS41-SGP | 1.0 hPa (> 100 hPa), 0.6 hPa (< 100 hPa) |
| | Temperature | | 0.3 °C (< 16 km) 0.4 °C (> 16 km) |
| | Relative humidity | | 4 % |
| | Wind speed | | 0.15 m s$^{-1}$ |
| | Wind Direction | | 2 ° |
| **Met Tower** | 2 m pressure | Vaisala PTU307 | 0.15 hPa |
| | 2 m wind speed | Metek uSonic-Cage MP sonic anemometer | 0.3 m s$^{-1}$ |
| **Ceilometer** | Cloud base height | Vaisala CL31 | 5 m |
| **Microwave radiometer** | Precipitable water vapor | Derived from ARM 2-channel microwave radiometer-measured brightness temperatures, in MWRRET Value-added Product | 0.3 mm |
| **Radiation station** | Longwave radiation | Eppley Precision Infrared Radiometer | 2.6 W m$^{-2}$ (downwelling) 1 W m$^{-2}$ (upwelling) |
| | Shortwave radiation | Eppley Standard Precision Pyranometer | 4.5 W m$^{-2}$ |

**2.2 Deriving quantities from observational data**

Before the radiosonde profiles were analyzed, radiosonde measurements were corrected to account for the local "heat island" resulting from the presence of the *Polarstern.* This local source of heat resulted in the frequent occurrence of elevated temperatures near the launch point, resulting in inconsistencies in the observed temperatures in the lowermost

part of the atmosphere. This phenomenon can be recognized by an artificial temperature structure indicative of a convective layer in the lowest radiosonde measurements, which we know is unlikely (Tjernström et al., 2004; Brooks et al., 2017). Thus, if this "convective layer" was present, then the lowest radiosonde measurements were visually compared to measurements from the met tower to identify when temperature values were anomalously warm. This was identifiable when the tower measurements interpolated upward, given their observed slope, did not match up with

the lowest radiosonde measurement. The first credible value of the radiosonde measurements was found when the tower measurements extrapolated upward would line up with the observed radiosonde measurement, or in the case of a temperature offset between the tower and radiosonde, would have the same slope (met tower measurements were not merged to the radiosonde measurements due to frequent temperature offsets which could occur as a result of the two platforms sampling a slightly different airmass, differences in surface state, differences in instrument

accuracy/uncertainty, etc.). All data at the altitudes below this first credible value were removed. This helps in also removing faulty wind measurements that occur as a result of flow distortion around the ship (Berry et al., 2001).

An additional disruption of the radiosonde measurements sometimes occurred because of the passage of the balloon through the ship's exhaust plume. When it was unambiguous that the radiosonde passed through the ship's plume (evident by a sharp increase and subsequent decrease in temperature, typically by ~0.5-1°C over a vertical distance of ~10-30 m, identified visually), these values were replaced by values resulting from interpolation between the closest credible values above and below the anomalous measurements, which were identified as the last point just before the increase and the first point just after the decrease in temperature values, to acquire a continuous profile of reliable temperatures. Lastly, we determined that 92% of profiles have credible measurements as low as 35 m AGL. To allow for a consistent bottom height for our analysis, we only consider profiles in which there is a good measurement at 35 m, and do not consider data at altitudes below 35 m. This altitude is a compromise between removing too much low altitude data or removing too many radiosonde profiles from analysis. After removing all profiles in which there is not trustworthy data as low as 35 m, we retain 1377 MOSAiC radiosonde profiles for analysis.

Following the methods of Jozef et al. (2022) and Jozef et al. (2023a), ABL height from each radiosonde profile was determined by identifying the first altitude in which the bulk Richardson number ($Ri_b$) exceeds a critical value of 0.5 and remains above the critical value for at least 20 consecutive meters. $Ri_b$ was calculated using the following equation from Stull (1988):

$$Ri_b(z) = \frac{\left(\frac{g}{\overline{\theta_v}}\right)\Delta\theta_v\,\Delta z}{\Delta u^2 + \Delta v^2} \tag{1}$$

where $g$ is acceleration due to gravity, $\overline{\theta_v}$ is the mean virtual potential temperature over the altitude range being considered, $z$ is altitude, $u$ is zonal wind speed, $v$ is meridional wind speed, and $\Delta$ represents the difference over the altitude range used to calculate $Ri_b$ throughout the profile. $Ri_b$ profiles were created by calculating $Ri_b$ across 30 m intervals in steps of 5 m (Jozef et al., 2023a). This method identifies the ABL height as the bottom of the elevated $\theta_v$ inversion (or the bottom of the layer of enhanced $\theta_v$ inversion strength) for moderately stable to near-neutral conditions, and at the top of the most stable layer for conditions with a strong surface-based $\theta_v$ inversion.

CBH and PWV associated with each radiosonde were identified as the average of the measurements within 30 minutes before the radiosonde launch. Cloud frequency was determined as the percentage of observations within 30 minutes before radiosonde launch in which cloud presence was recorded. We used this 30 minute interval before the radiosonde observation, as this is a long enough time for the presence of the cloud and atmospheric moisture to impact atmospheric stability and structure close to the surface. Mixing ratio at ABL height was derived from the radiosonde profile, and the presence of fog was identified when the onboard meteorological observation closest in time to a given radiosonde launch reported fog.

Any other point measurements associated with each radiosonde (2 m wind speed and pressure, surface radiation budget components) were calculated as the average over a period of 5 minutes before to 5 minutes after radiosonde launch. The variables described in this section will hereafter collectively be called "composite variables."

**2.3 Stability regime analysis**

Twelve stability regimes have been defined based on stability within the ABL (hereafter referred to as "near-surface" stability) as well as the strength of the capping $\theta_v$ inversion located between the top of the ABL and 1 km (hereafter referred to as stability "aloft"; Table 2). By defining twelve distinct stability regimes, we expand upon the traditional categorization of stability into one of three categories: stable, neutral, and unstable (Stull, 1988; Liu and Liang, 2010). While some prior studies have separated the stable regime into a few subcategories for the Arctic (weakly stable, very

stable, and extremely stable; Sorbjan, 2010; Sorbjan and Grachev, 2010), our analysis expands upon this through the inclusion of additional subcategories for stability above the ABL. The stability regimes are used as classification bins for composite variables described in Sect. 2.2, for analysis of their variability with stability, and stability variability with season. Seasons are defined by grouping observations during September, October, and November (fall); December, January, and February (winter); March, April, and May (spring); and June, July, and August (summer).

The stability regime definitions were developed based on the results of a self-organizing map (SOM) analysis (which objectively identifies a user-selected number of patterns present in a training data set) conducted with the MOSAiC radiosonde profiles to reveal the range of vertical structures observed during MOSAiC (differentiated by stability within the ABL and the height and strength of a capping inversion) presented in Jozef et al. (2023b). The SOM revealed stability within the ABL to range from strongly stable to near-neutral and the stability aloft to range from strongly to

weakly stable.

The first step in identifying stability regime is calculating a virtual potential temperature gradient ($d\theta_v/dz$) profile. Since the stability criteria in part depend on stability within the ABL and some observations have an ABL height as low as 50 m, we first include a measurement of $d\theta_v/dz$ at 42.5 m (this determines the near-surface stability), calculated across a 15 m interval between 35 m (lowest point of the profile) and 50 m. For values at and above 50 m, $d\theta_v/dz$ is

calculated across 30 m intervals in steps of 5 m and attributed to the center altitude of $\Delta z$ (i.e., 35-65 m, 40-70 m, 45-75 m and so on), resulting in a $d\theta_v/dz$ profile with values at 42.5 m, 50 m, 55 m, 60 m AGL, and so on.

Table 2 shows the thresholds associated with each stability regime, and how they are applied. The first step for stability regime identification is to classify the near-surface stability using the $d\theta_v/dz$ value at 42.5 m. As the ABL at any given location is defined by the stability near the surface (Stull, 1988), this $d\theta_v/dz$ value at 42.5 m reasonably indicates the

ABL stability. The possible near-surface regimes are strongly stable (SS), moderately stable (MS), weakly stable (WS) and near-neutral (NN). Near-surface instability is not considered as its own category, as the instances are very few, and any such cases are grouped into the NN category. To differentiate between stable cases (SS, MS, or WS) and near-neutral cases (NN), we use a threshold of 0.5 K $(100 \text{ m})^{-1}$, where if $d\theta_v/dz$ below 50 m is less than the threshold, it is considered NN, and if it is greater than or equal to the threshold, it is stable. This threshold was chosen, as it equates

to the threshold of 0.2 K over 40 m used to discern a stable versus neutral ABL in Jozef et al. (2022), adapted from thresholds given in Liu and Liang (2010). Additional thresholds were derived to differentiate SS, MS, and WS. While a range of thresholds were tested, the ones listed in Table 2 were determined to best discern meaningful differences in

near-surface $\theta_v$ inversion strength for both the MOSAiC data presented here as well as radiosonde profiles at several sites in Antarctica (Dice et al., submitted).

The second step for stability regime identification is only applied to cases with a near-surface regime of WS or NN and is carried out to differentiate weakly stable or near-neutral cases (both considered relatively well-mixed) that are very shallow, from those that are deeper. We make this distinction because there are different processes that would lead to a shallow versus deep well-mixed layer. Thus, if ABL height is less than 125 m, we consider this a very shallow mixed (VSM) case. This threshold of 125 m was chosen, as there is a cluster of SOM patterns in Jozef et al. (2023b)

with near-surface regime of WS or NN that have ABL height less than 125 m, and a jump in height before the next cluster of SOM patterns with ABL height above 125 m. The ABL height is not relevant for the definition of SS and MS, though these regimes usually have an ABL height less than 125 m, and SS cases often have an ABL height as low as 50 m.

Lastly, stability aloft is determined. This step is only applied to VSM, WS, and NN cases, as we only address stability

aloft if it is more stable than the near-surface stability regime. For SS and MS cases, the profile is at its most stable near the surface, and transitions to the free atmosphere above the ABL, so stability aloft does not provide additional information. Using the maximum in the $d\theta_v/dz$ profile above the ABL, but below 1 km, the same thresholds as previously applied to identify the near-surface regime are also applied to identify stability aloft, where the options are strongly stable aloft (SSA), moderately stable aloft (MSA), and weakly stable aloft (WSA).

While other studies define stability in the Arctic based on $Ri_b$ and local Obukhov length (Sorbjan, 2010; Sorbjan and Grachev, 2010), or based on temperature lapse rate (Pithan et al., 2014), we found the above methods for defining stability regime based on $d\theta_v/dz$ and ABL height to yield reliable results while providing the best potential for repeatability in future work (e.g., Dice et al., submitted), as the methods rely only on standard radiosonde observations (and do not require additional measurements). This also allows us to apply the same methods to both the near-surface

and aloft stabilities. Additionally, as the focus of this study is to analyze the relationships between turbulent forcing mechanisms and stability, metrics for stability regime identification that include these forcing mechanisms in their definition (e.g., Obukhov length and $Ri_b$ include wind speed in their calculations) were avoided. Comparison of the stability regimes determined using the methods described in this section to bulk friction velocity from the met tower (Jozef et al., 2023b) shows that the current methods discern meaningful differences in turbulence between the various

stability regimes.

All of the resulting options for stability regime are listed in Table 2, and an example case for each regime (except NN) is shown in Fig. 2. The color-coding in Table 2 will be used to discern each regime henceforth. While we list NN as a stability regime option, a purely NN case without enhanced stability aloft was never observed in a MOSAiC radiosonde profile, and as such no NN example is given in Fig. 2.


**Table 2:** Thresholds used to differentiate between stability regime, where the various near-surface regimes are SS (strongly stable), MS (moderately stable), VSM (very shallow mixed), WS (weakly stable) and NN (near-neutral), and the various stabilities aloft are SSA (strongly stable aloft), MSA (moderately stable aloft), and WSA (weakly stable aloft).

| $d\theta_v/dz$ at 42.5 m AGL | ABL Height | Max. $d\theta_v/dz$ above ABL | Stability Regime | Abbreviation |
|---|---|---|---|---|
| $\geq 5$ K (100 m)$^{-1}$ | - | - | Strongly Stable | SS |
| $\geq 1.75$ K (100 m)$^{-1}$ $< 5$ K (100 m)$^{-1}$ | - | - | Moderately Stable | MS |
| $< 1.75$ K (100 m)$^{-1}$ | $< 125$ m | $\geq 5$ K (100 m)$^{-1}$ | Very Shallow Mixed – Strongly Stable Aloft | VSM-SSA |
| | | $\geq 1.75$ K (100 m)$^{-1}$ $< 5$ K (100 m)$^{-1}$ | Very Shallow Mixed – Moderately Stable Aloft | VSM-MSA |
| | | $< 1.75$ K (100 m)$^{-1}$ | Very Shallow Mixed – Weakly Stable Aloft | VSM-WSA |
| $\geq 0.5$ K (100 m)$^{-1}$ $< 1.75$ K (100 m)$^{-1}$ | $\geq 125$ m | $\geq 5$ K (100 m)$^{-1}$ | Weakly Stable – Strongly Stable Aloft | WS-SSA |
| | | $\geq 1.75$ K (100 m)$^{-1}$ $< 5$ K (100 m)$^{-1}$ | Weakly Stable – Moderately Stable Aloft | WS-MSA |
| | | $< 1.75$ K (100 m)$^{-1}$ | Weakly Stable | WS |
| $< 0.5$ K (100 m)$^{-1}$ | | $\geq 5$ K (100 m)$^{-1}$ | Near-Neutral – Strongly Stable Aloft | NN-SSA |
| | | $\geq 1.75$ K (100 m)$^{-1}$ $< 5$ K (100 m)$^{-1}$ | Near-Neutral – Moderately Stable Aloft | NN-MSA |
| | | $\geq 0.5$ K (100 m)$^{-1}$ $< 1.75$ K (100 m)$^{-1}$ | Near-Neutral – Weakly Stable Aloft | NN-WSA |
| | | $< 0.5$ K (100 m)$^{-1}$ | Near-Neutral | NN |


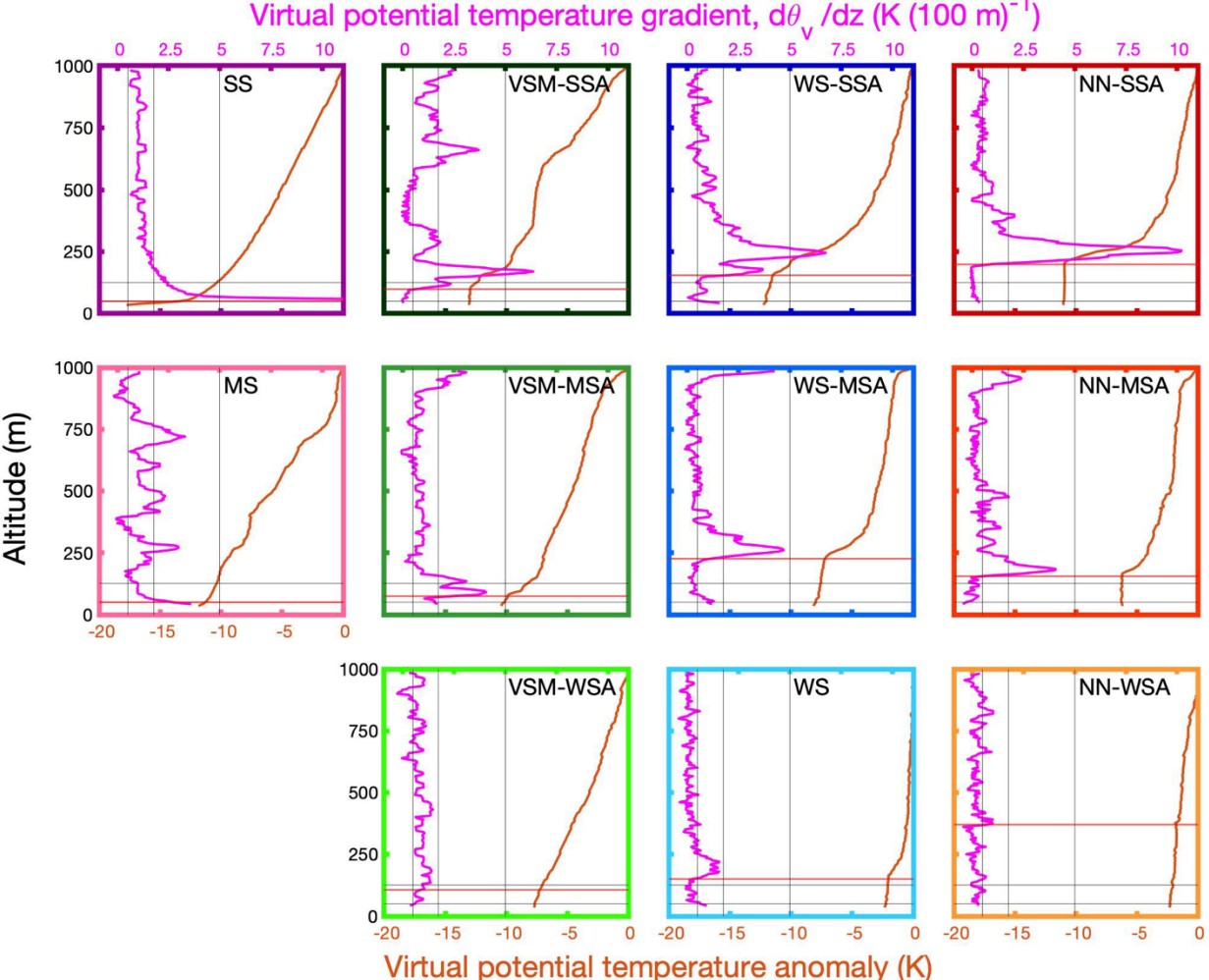

**Figure 2.** Example cases for each stability regime listed in Table 2, except NN, showing profiles of virtual potential temperature ($\theta_v$) anomaly with respect to 1 km (orange line, bottom x-axis) and virtual potential temperature gradient ($d\theta_v/dz$; magenta line, top x-axis). Vertical black lines (at 0.5, 1.75, and 5 K (100 m)$^{-1}$) and horizontal black lines (at 50 and 125 m AGL) in each subplot indicate the various thresholds used to determine stability regime. The horizontal red line in each subplot is the ABL height for that example. Stability regime of the example is written on each subplot and is also indicated by the color of the border.

## 3 Results and discussion

### 3.1 Frequency of stability regimes

Annual and seasonal frequencies of ABL stability regimes, based on all radiosonde observations during MOSAiC, is shown in Fig. 3. For the stability regime frequencies shown in Fig. 3 and subsequent figures, the regimes are organized from strongest to weakest near-surface stability going from left to right (where VSM is considered more stable than WS due to a shallower ABL), and within a given near-surface regime, the aloft regimes are also organized such that stability decreases from left to right.

Annually, the stability regime which occurred with the highest frequency was NN-SSA followed by VSM-SSA. In decreasing order, MS, SS, and NN-MSA, also occurred with high frequency. VSM-MSA occurred with moderate frequency, and VSM-WSA, NN-WSA, and all WS regimes were relatively infrequent. The high frequency of regimes with either moderate or strong stability near the surface, or a well-mixed ABL with strong stability aloft, suggests that the central Arctic lower atmosphere tends towards being strongly stable, but sometimes the near-surface atmosphere can become well-mixed due to the generation of turbulence.

In fall, the strongest stability regimes (SS and MS) were less frequent, while NN near the surface was more frequent. This may be due to the thinner sea ice which results in more upward heat transfer from the ocean to the atmosphere, but more likely is because autumn is characterized by the highest frequency of low-level liquid-bearing clouds (Shupe et al., 2011a; Shupe et al., 2011b), which contributes to the weakening of near-surface stability. Of all seasons, the winter stability regime frequency distribution was most different from the annual results. Winter had a higher frequency of the strongest stability regimes (SS, MS, and VSM-SSA), and the NN near-surface regime was heavily dominated by NN-SSA. Thus, there was a clear dominance of stronger stability in winter compared to other seasons, which is expected due to the lack of solar radiation and corresponding dominance of longwave cooling of the surface, which promotes near-surface stability. In spring, the relative frequencies of stability regime were similar to the pattern that was seen annually. Lastly, in summer, the relative frequencies of SS, MS, VSM-SSA, VSM-MSA, NN-SSA, and NN-MSA were very similar to one another, which suggests that the forcing mechanisms of each of these regimes occurred with similar frequency, or that certain regimes may occur under a range of forcing mechanisms.

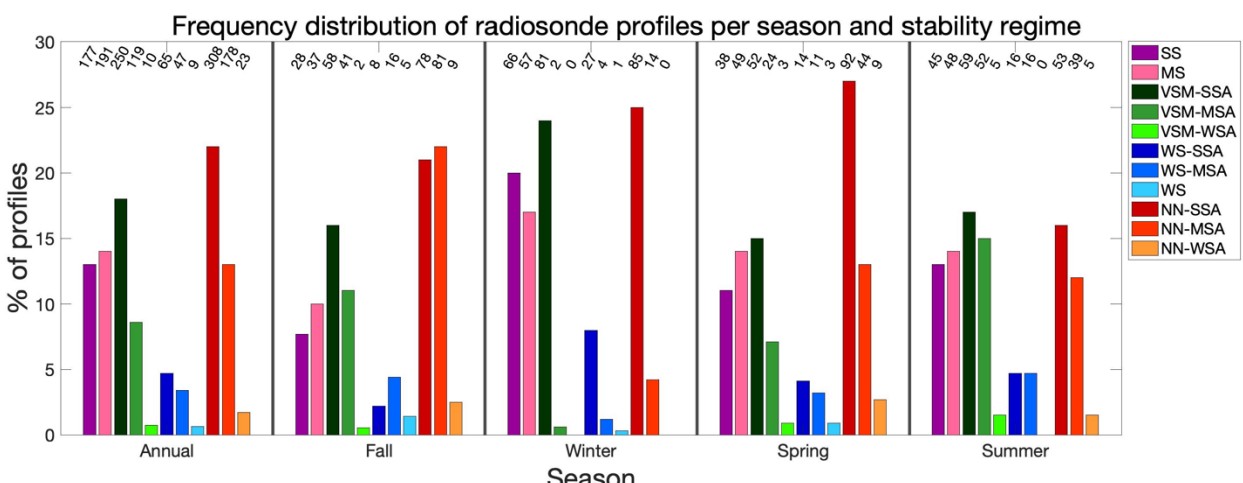

**Figure 3.** Frequency distribution showing the percent of radiosonde profiles in each stability regime, annually and seasonally. For the seasonal sections, the percent shown is with respect to the total number of radiosonde profiles in that season. The numbers along the top of the plot, above each bar, indicate the total number of radiosonde profiles of that stability regime and season.

**3.2 Stability regime transitions**

Knowing how the various stability regimes transitioned between each other helps to understand how and why the regimes may form. Figure 4 shows, for each observation of a given stability regime (rows), the frequency with which

the other regime options occurred in the prior observation (normally 6 h earlier in accordance with the sounding schedule) (columns). The values across the diagonal from the upper lefthand corner to the lower righthand corner

indicate when the same regime occurred in the previous observation as was present in the current observation. Unsurprisingly, for most stability regimes, the one that occurred most often previously was the same stability regime. The stability regime that had the highest frequency of the same regime (persistence) in the previous observation is SS, followed by NN-SSA, which suggests that strong forcings are necessary to change these regimes. This is further supported by the fact that the SS and NN-SSA regimes had the largest and second largest observed number of

consecutive cases, respectively (there was one instance of 12 consecutive SS cases (~66 hours), and one instance of 10 consecutive NN-SSA cases (~54 hours)). These two regimes also had a higher occurrence of persisting for at least three observations ($\geq$ 12 hours) than all other regimes. VSM-SSA had the next highest number of occurrences of persisting for at least three observations.

Aside from itself, SS largely only occurred after the MS or VSM regimes, with less than 5% of SS cases occurring

directly after a WS or NN case, which means that SS conditions generally only form when the ABL is already shallow and the surface-based or near-surface inversion is already relatively strong. MS most frequently occurred following SS and VSM-SSA, aside from itself, for the same reasoning as discussed for the SS regime. Aside from themselves, VSM-SSA and WS-SSA most frequently occurred after NN-SSA, and NN-SSA most frequently occurred after VSM-SSA. Thus, we conclude that when there is strong stability aloft, it is likely to persist, but the depth and stability within

the ABL may still be altered as a result of mechanically or radiatively driven turbulent forcings. Aside from themselves, VSM-MSA most frequently occurred after VSM-SSA followed by NN-MSA, WS-MSA most frequently occurred after NN-SSA followed by NN-MSA, and NN-MSA most frequently occurred after NN-SSA followed by VSM-MSA. Thus, when there was moderate stability aloft, there was less consistency in the stability which occurred before, so this moderate stability aloft is less likely to persist than strong stability aloft. The same is true for weak

stability aloft and near the surface. This leads to the conclusion that the central Arctic lower atmosphere is inclined to be strongly stable somewhere in the lowest 1 km, but the height of this strongly stable layer can become elevated, separated from the surface by a well-mixed layer, when turbulence is generated. This additionally leads to the conclusion that moderate and weak stability aloft, as well as weak stability near the surface, likely are representative of transitional states (e.g., perhaps between clear and cloudy states).

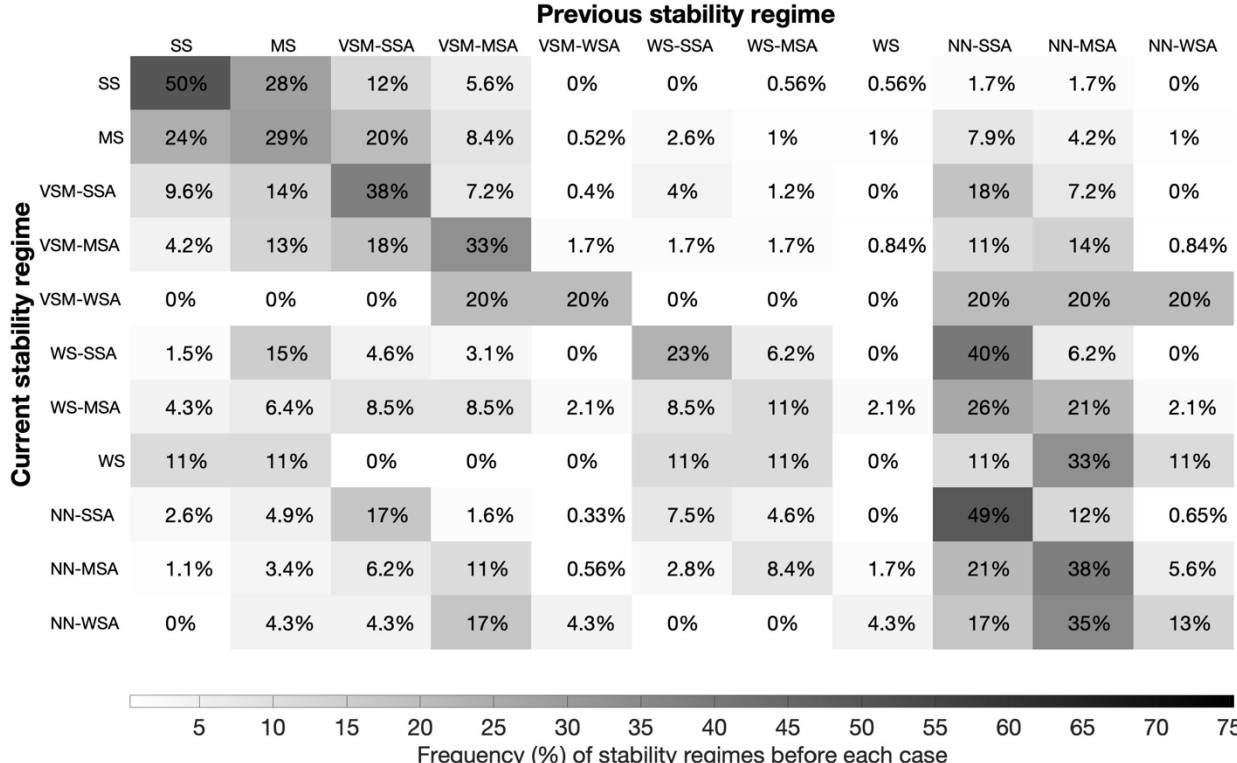

**Figure 4.** Grid plot showing, for each stability regime, what was the frequency of the previous case's stability regime, where the rows indicate the current stability regime, and the columns indicate the stability regime of the previous radiosonde observation. The greyscale color bar corresponds to the percent of previous cases in each stability regime, where darker grey signifies a higher percent of cases.

### 3.3 Mechanical impact on stability regime

For the remainder of the paper, we are going to look at how near-surface wind speed and surface radiative fluxes correspond to stability regime. We approach this analysis with the understanding that both wind and radiation can produce turbulence (as discussed in Sect. 1), and thus we assume that the observed stability regimes largely occurred as a response to the observed wind and radiation features (e.g., enhanced wind speeds and radiation work to weaken ABL stability through mechanically and thermodynamically generated turbulence). However, we do recognize the possibility that the observed wind and radiation features could have occurred as a response to the observed stability, and in many cases, it may be a combination of interactions in both directions.

Wind speed at 2 m is used as a proxy for the amount of near-surface wind shear, and subsequent mechanical mixing, impacting ABL stability, depicted by Fig. 5a-e, which shows the range of 2 m wind speed for each stability regime and season. Supplementary Fig. S1 indicates when there is a statistically significant ($p < 0.05$) difference in the mean values of 2 m wind speed between all pairs of stability regimes, using a two-tailed t-test when degrees of freedom (df) $\leq 100$, and a two-tailed z-test when df $> 100$. Annually (Fig. 5a), the mean and median values of 2 m wind speed were 4.5 and 4.1 m s$^{-1}$ respectively, and as 2 m wind speed increases, near-surface stability decreases, indicating that wind speed is correlated to ABL stability. This agrees with the well-documented notion that stability is dependent on wind speed (Brooks et al., 2017; Banta et al., 2003), as is reflected in the definition of Ri$_b$ (Stull, 1988), which is often used

as a metric for determining stability. As stability regime classification in the current study is not directly dependent on wind speed, evidence for this relationship is strengthened. There is a step change increase in 2 m wind speed from SS, MS, and the VSM regimes (mean of 3.0 m s$^{-1}$) to the WS and NN regimes (mean of 6.3 m s$^{-1}$), where SS, MS, and the VSM regimes largely had 2 m wind speed below average, and the WS and NN regimes largely had 2 m wind speeds above average. Thus, faster wind speeds likely contribute to mechanical mixing near the surface that works to weaken near-surface stability and deepen the ABL, leading to a WS or NN case. This is supported by Supplementary Fig. S1a which shows a significant difference in 2 m wind speed when comparing SS to all other regimes, and when comparing MS and the VSM regimes to the WS and NN regimes. However, there is little significance when comparing MS and the VSM regimes to each other, or when comparing the WS and NN regimes to each other. Within the near-surface regimes with varying stability aloft (VSM, WS, and NN), 2 m wind speed decreases as stability aloft decreases, which suggests that when stability aloft is stronger, more mechanically generated turbulence, and thus faster near-surface wind speeds, are necessary to mix out the near-surface layer.

Seasonally, there was little difference from the annual pattern, however there are some notable discrepancies. In winter (Fig. 5c), there is a larger increase in 2 m wind speed between SS, MS and the VSM regimes and the WS and NN regimes (increase of 3.7 m s$^{-1}$ versus 3.3 m s$^{-1}$ annually), and a greater number of regimes that have significantly different values from each other (Fig. S1c), suggesting that near-surface wind speed is a more important driver of ABL stability in winter than the other seasons. In summer, there is a smaller increase in 2 m wind speed between SS, MS and the VSM regimes and the WS and NN regimes (increase of 2.7 m s$^{-1}$ versus 3.3 m s$^{-1}$ annually; Fig. 5e), but there is still high significance (Fig. S1c) in the difference between the stronger stability regimes (SS, MS and VSM) to the weaker stability regimes (WS and NN). Thus, while in summer wind shear may not be the most important variable differentiating stability, it still plays a significant role.

One potential explanation for differences in wind speed in the central Arctic is the synoptic setting, which can be inferred with the 2 m pressure and pressure tendency (near-surface pressure may also be linked to surface longwave radiative flux where lower surface pressure (i.e., a storm) corresponds to higher longwave radiation (i.e., cloudy state); Morrison et al., 2012). Figure 5f-j shows the range of 2 m pressure and Fig. 5k-o shows the range of absolute 2 m pressure tendency (dp/dt) corresponding to each radiosonde launch for each stability regime and season (refer to Supplementary Fig. S2 for corresponding significance testing), where the annual mean of 2 m pressure and dp/dt throughout MOSAiC were 1010.8 hPa and 0.77 hPa (3 hr)$^{-1}$ respectively. Annually, the pressure results mimic what was seen with 2 m wind speed, in that lower pressure and greater dp/dt (suggestive of a stormy setting with faster wind speeds) is correlated with weaker stability, with the most drastic reduction in pressure and increase in dp/dt values being between SS, MS and the VSM regimes (pressure largely above average and dp/dt largely below average) and the WS and NN regimes (pressure largely below average and dp/dt largely above average; difference in means of 6.6 hPa and 0.31 hPa (3 hr)$^{-1}$ respectively). This is supported by Supplementary Fig. S2a which shows a high level of significance when comparing 2 m pressure and dp/dt between different stability regimes. Through this process, it is possible that high wind speeds associated with a storm could change the surface roughness (e.g., a storm causes sea ice movement and the subsequent formation of ridges) which then impacts how turbulence production is influenced

by the surface under a given wind regime. Thus, it is possible that high wind speed events could have repercussions that contribute to weakening of ABL stability not only during the wind event, but also through increased surface roughness afterwards. However, this is a theory that would need further testing and is outside the scope of the current study.

The seasonal 2 m pressure results follow a similar trend, with the largest difference in 2 m pressure between stability regimes, and the most significant differences occurring in winter (Fig. 5h and S2c), which echoes the results found from the 2 m wind speed. This suggests that synoptic scale storms are a major factor leading to an increase in near-surface wind speeds which contribute to weak or near-neutral stability in winter. Differences in 2 m dp/dt between stability regimes in winter are not as great as annually or in fall or spring (Fig. 5m and S2c) suggesting more slowly evolving low and high pressure systems in winter than in other seasons. The smallest differences in 2 m pressure and dp/dt between stability regimes occurred in summer (Fig. 5j and S2e), again echoing the results from the 2 m wind speed, and further supporting the statement that the presence of storms, and resulting wind shear, are not the most important drivers of ABL stability in summer.

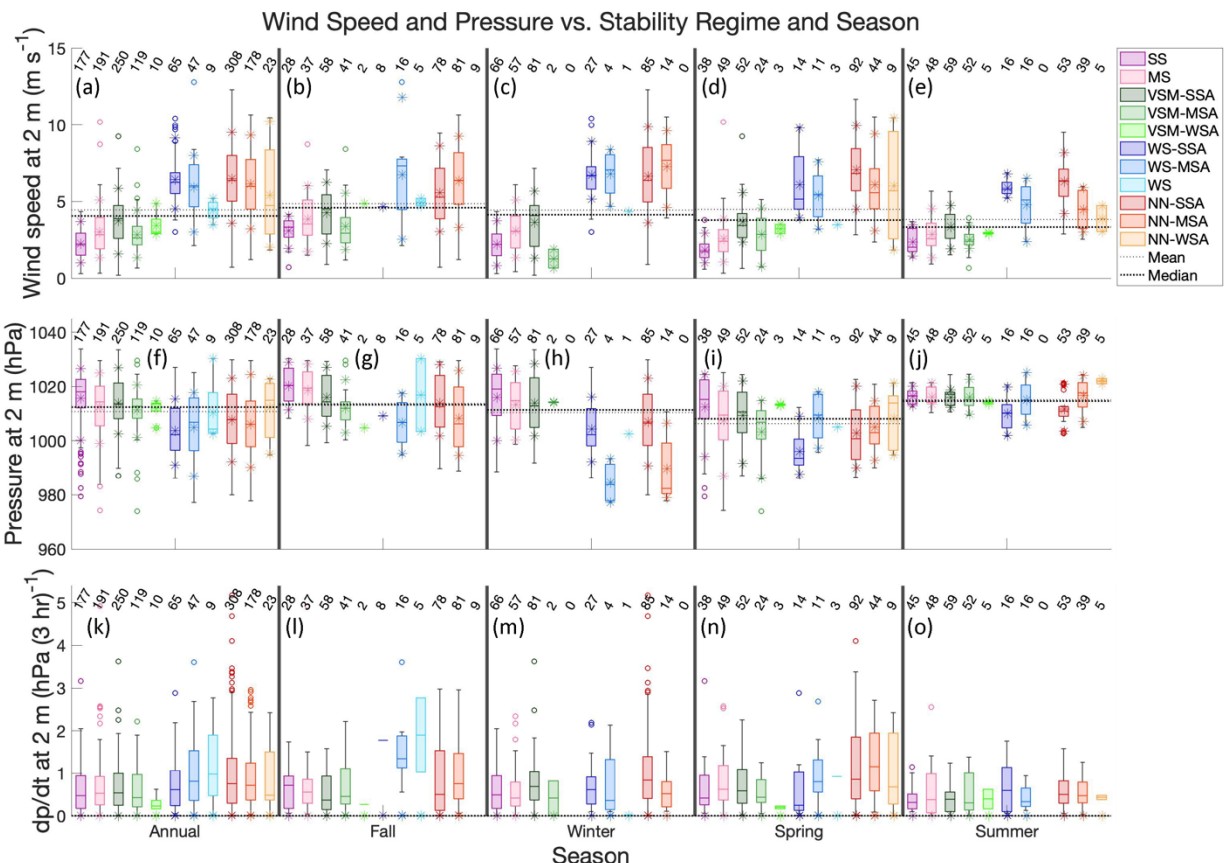

**Figure 5.** Top: Box and whisker plots showing the range of 2 m wind speed (a) annually, and during (b) fall, (c) winter, (d) spring, and (e) summer for each stability regime. Middle: Box and whisker plots showing the range of 2 m pressure (f) annually, and during (g) fall, (h) winter, (i) spring, and (j) summer for each stability regime. Bottom: Box and whisker plots showing the range of 2 m pressure tendency (k) annually, and during (l) fall, (m) winter, (n) spring, and (o) summer for each stability regime. The center line of each box is the median, and the outer ranges of the boxes

are the upper and lower quartiles. The whiskers show the range of values within 1.5 times the interquartile range from the top or bottom of the box, and outliers are shown with hollow circles. Asterisks are included at the mean, 10th percentile, and 90th percentile. Horizontal dotted black lines show the annual and seasonal mean (light dotted) and median (heavy dotted) values of each variable. The number of cases in each stability regime are written along the top of the figure.

**3.4 Radiative impact on stability regime**

The downwelling components of the surface radiation budget indicate the atmospheric forcing on the radiative production of turbulence and the subsequent impact on ABL stability. These feed into the net radiation experienced at the surface, which can be analyzed to determine when surface melt is possible (net radiation > 0). Thus, Fig. 6 shows the range of net radiation (Fig. 6a-e) as well as the downwelling longwave (Fig. 6f-j) and shortwave (Fig. 6k-o) components for each stability regime and season (refer to Supplementary Fig. S3 and S4 for corresponding significance testing).

Annually, the mean and median of net radiation were -21.1 and -25.8 W m$^{-2}$, indicating that over the course of the MOSAiC year, the radiative balance at the *Polarstern* was negative (i.e., there was surface cooling). Net radiation values were highest for the VSM and NN regimes (which had net radiation largely above average), whereas SS, MS, and the WS regimes (which had net radiation largely below average) had similar, lower values. There is a significant difference between net radiation for most regimes, however largely a lack of significance within the WS regimes, and between the VSM and NN regimes. This suggests that the VSM and NN regimes similarly occur under higher net radiation conditions (i.e., less negative and sometimes positive), but under these high radiation conditions there is an additional factor that dictates whether stability develops into VSM or instead develops into NN. Based on the results discussed in Sect. 3.3, this additional factor is likely wind speed, with stronger winds leading to the deeper ABL NN regimes rather than shallower ABL VSM regimes. Additionally, only the 75th percentiles (upper limit of the interquartile range) of net radiation for the VSM-MSA, VSM-WSA, and NN-WSA regimes exceed zero (Fig. 6a), and thus surface melt is more likely for these stability regimes. The seasonal trends were largely similar to the annual trend, aside from summer, for which there is no significant difference in net radiation between any two stability regimes. The summer radiation conditions and the connection to stability regime will be discussed in detail below. Aside from summer, conditions for surface melt are rare, but were more common in spring than in fall or winter.

Downwelling longwave radiation, with variability driven primarily by cloud cover and cloud temperatures (as discussed below) had an annual mean and median of 196.0 and 181.2 W m$^{-2}$ respectively (Fig. 6f). A similar trend as was seen in the net radiation is also seen in the downwelling longwave component, in that the VSM and NN regimes had the highest values, largely above average. SS and MS had the lowest values, largely below average, and the WS regimes had downwelling longwave radiation values somewhere in between, closer to the average. Comparison of surface net longwave radiation to ABL stability reveals that there is a bimodal distribution with weaker stability more often occurring in the cloudy sky mode (surface let longwave greater than -25 W m2) and stronger stability more often occurring in the clear sky mode (surface let longwave less than -25 W m2). Further, within the clear sky mode, stronger stability corresponds to weaker longwave cooling. These results agree with Pithan et al. (2014) which revealed these

conclusions using data from the SHEBA project. The seasonal trends in downwelling longwave radiation are largely similar to what was seen annually, aside from summer, which had similar (high) values of downwelling longwave radiation for all regimes, and will be discussed later on. In winter, there were overall lower values of downwelling longwave radiation due to the colder temperatures.

The annual values of downwelling shortwave radiation (Fig. 6k) are less useful, as they are heavily impacted by zero values for much of the year (in winter and some of the time in fall), though the VSM and NN regimes still had the highest annual values of downwelling shortwave radiation among all regimes (the validity of this signal in the annual results is supported by the spring observations). In spring, similarly to net radiation, SS, MS, and the WS regimes had similar (lower) values, and the VSM and NN regimes had higher values, further supporting that VSM and NN are radiatively driven. Again, the trend in downwelling shortwave radiation in summer is different than what was observed during the other seasons.

There are a greater number of regimes in which downwelling radiation is significantly different from the other regimes for longwave versus shortwave radiation (Fig. S4a). This all suggests that longwave radiation is more coupled to ABL stability throughout the span of the year than shortwave radiation. Within the near-surface regimes that have enhanced stability aloft (VSM, WS, and NN), for all radiation variables being considered, there is an increase in stability aloft with decreasing radiation. This suggests that radiation is also connected to stability aloft.

The characteristics of the surface radiation budget and its relationship with ABL stability differed in summer from what was observed throughout the rest of the year. Net radiation exceeded zero for all regimes (Fig. 6e), consistent with the fact that the surface of the Arctic sea ice experiences melt in the summer. However, it is perhaps counterintuitive that there was positive net radiation for the SS and MS regimes because strong stability usually occurs due to radiative cooling of the surface, which leads to a surface-based inversion. Thus, in summer, it is likely an advective process, rather than radiative cooling, that results in stronger near-surface stability. This advective process usually manifests in warm moist air advection from over the relatively warmer open ocean (of which there is more in summer) to over the relatively colder sea ice surface (whose temperature will be fixed at 0 °C over a melting ice surface), which decouples the colder near-surface atmosphere from the advected layer, resulting in a shallow surface-based inversion and stable ABL. A common signature of this process is fog. Thus, this warm air advection and resulting fog likely explains the trends in downwelling radiation observed in summer.

For example, fog, which is optically thin compared to a typical low-level cloud, lets through more shortwave radiation than a low cloud does, but would also produce large amounts of downwelling longwave radiation due to the high moisture content. This may explain why, for the SS regime, we saw the highest values of downwelling shortwave radiation (Fig. 6o), as well as similarly high values of downwelling longwave radiation (Fig. 6j) as the other stability regimes in summer. Thus, in summer, a strong and moderately stable ABL can occur under similar radiative conditions as would result in a VSM or NN regime at any other time of year, and therefore the net radiation values as well as the downwelling longwave and shortwave radiation components for all regimes are very similar, or even decrease with decreasing stability in the case of net and downwelling shortwave radiation. This is further supported by Fig. S3c and

Fig. S4e which show that very few stability regimes are significantly different from each other with regards to radiation in summer.

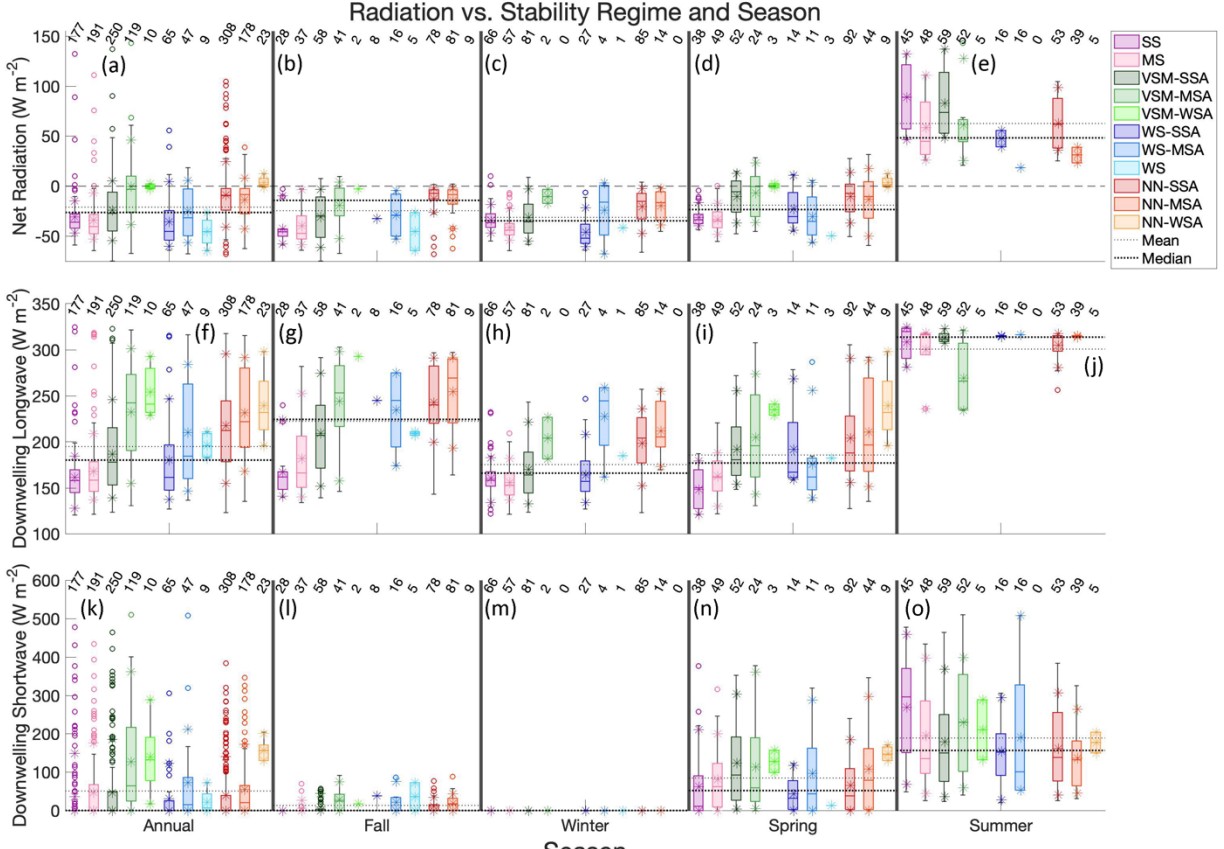

**Figure 6.** Top: Box and whisker plots showing the range of net radiation (a) annually, and during (b) fall, (c) winter, (d) spring, (e) summer for each stability regime. Middle: Box and whisker plots showing the range of downwelling longwave radiation (f) annually, and during (g) fall, (h) winter, (i) spring, and (j) summer for each stability regime. Bottom: Box and whisker plots showing the range of downwelling shortwave radiation (k) annually, and during (l) fall, (m) winter, (n) spring, and (o) summer for each stability regime. The center line of each box is the median, and the outer ranges of the boxes are the upper and lower quartiles. The whiskers show the range of values within 1.5 times the interquartile range from the top or bottom of the box, and outliers are shown with hollow circles. Asterisks are included at the mean, 10th percentile, and 90th percentile. Horizontal dotted black lines show the annual and seasonal mean (light dotted) and median (heavy dotted) values of each variable. The number of cases in each stability regime and season are written along the top of the figure.

The presence of clouds helps to explain the surface radiation characteristics seen in Fig. 6. Thus, the frequency of total cloud cover as well as only low cloud cover (CBH <= 2 km) within 30 minutes prior to each radiosonde launch (percent of ceilometer observations in the 30 minute window which contained clouds) are shown in Fig. 7a (refer to Supplementary Fig. S5 for corresponding significance testing). Only the mean values for each stability regime, as well as the overall annual and seasonal means are plotted because the range of values for each stability regime is wide, and thus the clearest differences between cloud frequency for the varying stability regimes can be seen with simply the mean. The annual mean of total cloud frequency was 49% and total cloud cover was dominated by low clouds (78% of clouds observed were low clouds), which had an annual mean frequency of 41%. The mean frequency of both total

clouds and low clouds was greatest in fall (68% and 59% respectively), likely due to the thinner and less extensive sea ice which results in more upward moisture transfer from the ocean to the atmosphere, consistent with Shupe et al. (2011b). The lowest frequency in total cloud cover was in winter and spring (40%), with summer having the lowest frequency in low cloud cover (25%).

Low clouds, which have a high moisture content, emit large amounts longwave radiation due to their high optical thickness and warm temperatures, and thus it is expected that low cloud frequency would mirror the trend in the annual downwelling longwave radiation with stability, as seen in Fig. 7a. Thus, clouds correspond to weakened ABL stability, manifesting in VSM or NN regimes occurring when there was a higher frequency of cloud cover. In these cases, the clouds were likely weakening the ABL stability both through warming of the surface by enhanced downwelling longwave radiation that leads to turbulence production, as well as through mixing below the cloud base through cloud top radiative cooling. Conversely, the SS and MS regimes largely occurred in the absence of clouds. This is supported by Fig. S5a which shows that SS and MS cloud frequencies are significantly different than those from nearly all other regimes, but not significantly different from each other. This again agrees with the results of Pithan et al. (2014) which showed that stability is stronger in the Arctic clear sky state.

Interestingly, while the relationship between radiation and stability was essentially the opposite in summer from what was observed during the other seasons, the relationship between cloud frequency and stability in summer was the same as what was observed throughout the rest of the year. However, due to the phenomenon (presented in previous literature, e.g., Tjernström, 2005 and Tjernström et al., 2019) of warm air advection from over the open ocean to over the sea ice leading to a stable ABL in summer, as indicated by the presence of a fog layer, we might expect a higher frequency of cloud cover for the SS and MS regime in summer. The likely reason we do not necessarily see this in Fig. 7e is because usually when cloud cover is contributing to the formation of an SS or MS ABL, it is in the form of fog, and since the ceilometer measuring cloud was situated on the deck of the *Polarstern*, it was sometimes above the fog layer, and thus did not always record the presence of a cloud. The presence of fog is better represented by the manual meteorological observations conducted from the *Polarstern*. The percent of radiosonde observations in summer during which fog was reported, coinciding with the varying stability regimes, is shown in Fig. 7f. Here, it is revealed that in fact the frequency of fog when SS or MS were observed is greater than the cloud frequency shown in Fig. 7e. Thus, the suggestion of warm air advection, identified using fog presence, contributing to a stable ABL in summer is supported. However, further work, such as airmass trajectory analysis, would be needed to fully prove this hypothesis.

In fall, winter, and spring, the relationship between cloud frequency and stability regime was similar to the annual trend, however, when analyzed seasonally, there are fewer pairs of stability regime in which cloud frequency is significantly different. This is particularly true in winter (Fig. 7c) which only has four instances of significant difference between regimes (Fig. S5c), likely arising simply due to the overall low cloud frequency throughout winter. Thus, while clouds are a good explanation for some of the radiative characteristics of the atmosphere, they do not

provide a full explanation for the relationships seen between the surface radiation components and ABL stability, and thus we turn to some other moisture variables for further explanation.

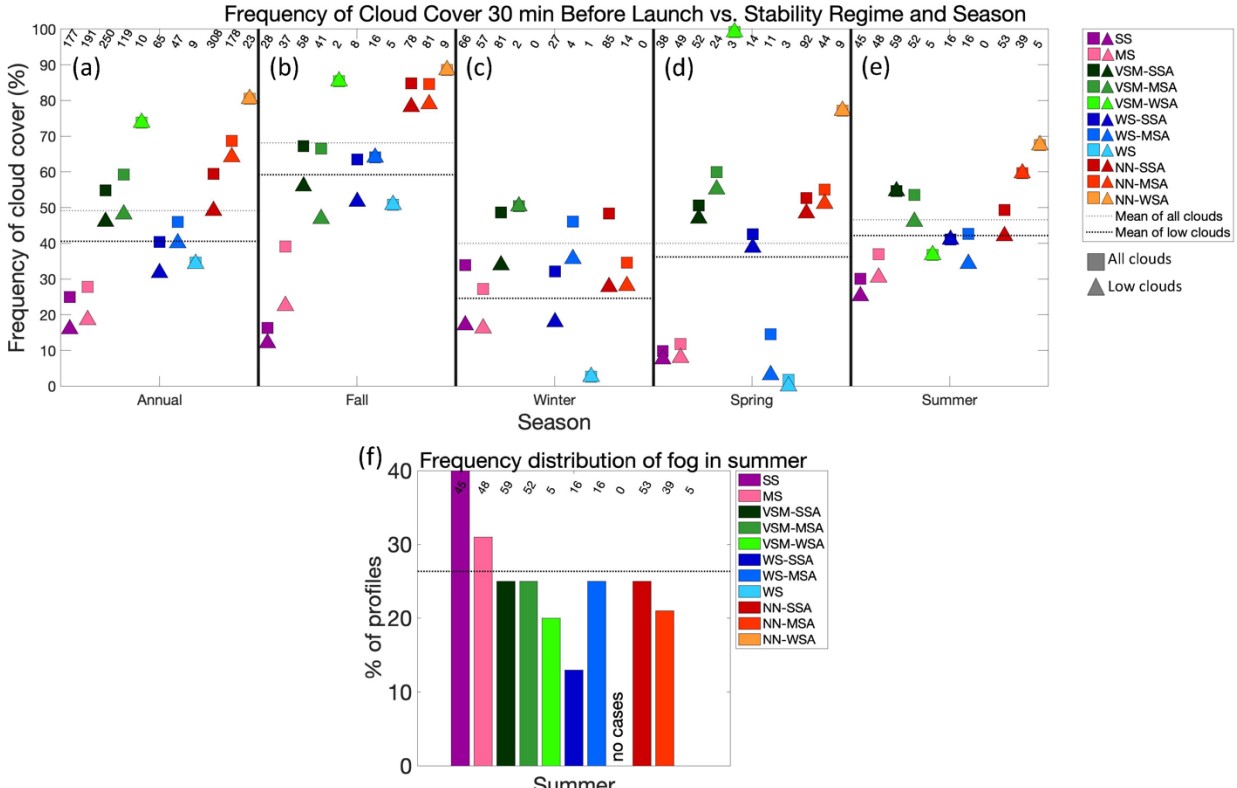

**Figure 7.** Top: Mean frequency of cloud cover within 30 minutes before radiosonde launch (a) annually, and during (b) fall, (c) winter, (d) spring and (e) summer for each stability regime. Square symbols show mean frequency of all clouds, and triangle symbols show mean frequency of low clouds only (cloud base height <= 2 km). Horizontal dotted black lines show the annual and seasonal mean values of frequency of all cloud cover (light dotted) and low cloud cover (heavy dotted). The number of cases in each stability regime and season are written along the top of the figure. Bottom: (f) Percent of radiosonde profiles during summer in which fog was present, depending on stability regime. Horizontal dotted black line indicates the overall frequency of fog in summer. The number of cases in each stability regime are written along the top of the figure.

To further understand the forcings on the surface radiation budget, particularly downwelling longwave radiation, we visualize the range of some additional moisture variables. Figure 8a-e shows mixing ratio at ABL height (i.e., the mixing ratio just below the elevated $\theta_v$ inversion or layer of enhanced $\theta_v$ inversion strength) in the context of stability regime and season, as this is a direct measure of the amount of moisture which impacts the near-surface ABL through its radiative signature. Additionally, as ABL height varies throughout time, PWV is also shown (Fig. 8f-j) to support the results seen for mixing ratio at ABL height and provide further evidence of the impact of atmospheric moisture on ABL stability (refer to Supplementary Fig. S6 for corresponding significance testing). The annual mean and median of mixing ratio at ABL height were 1.78 and 1.15 g kg$^{-1}$ respectively, and the annual mean and median of PWV were 0.69 and 0.55 cm respectively. However, the signal is dampened in the annual quantities of these variables because the opposite relationship between atmospheric moisture and stability was observed in summer versus the other seasons.

In fall and spring, as mixing ratio at ABL height (Fig. 8b, d) and PWV (Fig. 8g, i) increase, stability largely decreases. Strangely, this does not fully correlate with the relationship between net and downwelling longwave radiation and stability, which showed the highest values for VSM and NN. In fact, the WS regimes, which were shown to occur in lower radiation environments, had the highest values for mixing ratio at ABL height and PWV. However, since cloud frequency for the WS regimes was lower than that for the VSM and NN regimes, this leads us to conclude that for the WS regimes, atmospheric moisture was concentrated closer to the surface and largely present in vapor form, rather than condensing into clouds (which have a greater radiative signature) at a higher altitude, as occurred more frequently for the VSM and NN regimes. As such, WS may be tied to subsidence. In both fall and spring, mixing ratio at ABL height is more significantly different between stability regimes than PWV (Fig. S6b and S6d), suggesting that the near-surface moisture influences stability more than the total amount of water in an atmospheric column. In winter, there were very low values of mixing ratio at ABL height and PWV, pointing to the extreme dry environment during Arctic winter, however the same general relationship between moisture and stability is true of winter as is true of fall and spring, again with more statistical significance in mixing ratio at ABL height between regimes (Fig. S6c).

The mixing ratio at ABL height and PWV help to further support the discussion that warm moist air advection leads to a strong ABL stability in summer, which is a phenomenon that is not seen in the other seasons. In summer (Fig. 8e and 9j), there was a similar relationship between stability and moisture for the VSM, WS and NN regimes as in the other seasons, but there were elevated moisture values for the SS and MS regimes in summer. This again is evidence for the idea of warm moist air advection driving the stronger stability regimes in summer. This is supported by supplementary Fig. S6e which shows SS and MS mixing ratio at ABL height to be significantly different than that for all other regimes. This is also true of PWV for SS versus all other regimes aside from MS, but MS PWV has less significant difference when compared to the other regimes. This again shows that near-surface moisture has a greater influence on ABL stability than the total amount of moisture in the atmospheric column. The same is true when comparing mixing ratio at ABL height to cloud frequency.

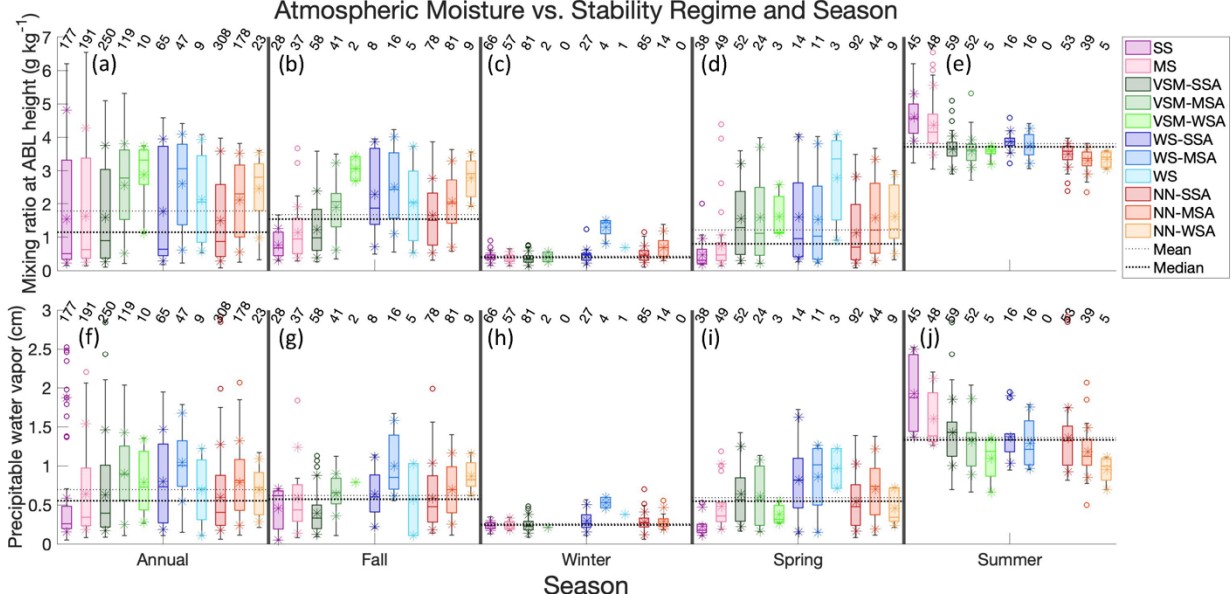

**Figure 8.** Top: Box and whisker plots showing the range of mixing ratio at ABL height (a) annually, and during (b) fall, (c) winter, (d) spring, and (e) summer for each stability regime. Bottom: Box and whisker plots showing the range of precipitable water vapor (f) annually, and during (g) fall, (h) winter, (i) spring, and (j) summer for each stability regime. The center line of each box is the median, and the outer ranges of the boxes are the upper and lower quartiles. The whiskers show the range of values within 1.5 times the interquartile range from the top or bottom of the box, and outliers are shown with hollow circles. Asterisks are included at the mean, 10th percentile, and 90th percentile. Horizontal dotted black lines show the annual and seasonal mean (light dotted) and median (heavy dotted) values of each variable. The number of cases in each stability regime and season are written along the top of the figure.

## 4 Summary and conclusions

The work presented in this paper provides a description of the seasonal frequency of ABL stability regimes, the interaction between thermodynamic and kinematic forcings and near-surface stability in the central Arctic, and how these relationships differ by season using data from the MOSAiC expedition. When grouping radiosonde observations by stability, it was determined that strong stability, either near the surface or aloft, was dominant (Fig. 3). The relative frequencies of stability regimes when separating the observations into the four seasons (fall, winter, spring, and summer) were most similar to the annual pattern for fall and spring. In winter, stronger stability was even more frequent. In summer, there were near equal frequencies of SS, MS, and VSM and NN with strong and moderate stability aloft, suggesting that these regimes may occur under a wide range of forcing mechanisms. By determining the frequency with which a certain stability regime occurred before each observation (Fig. 4), it was discovered that near-surface strong stability was most persistent (with one instance of SS persisting for ~66 hours), and moderate and weak stability aloft were less likely to persist than strong stability aloft (with NN-SSA and VSM-SSA having the greatest number of instances of persisting for at least 12 hours, compared to the other near-surface regimes with enhanced stability aloft), and thus moderate and weak stability aloft likely represent transition states. This leads to the conclusion that the central Arctic lower atmosphere is inclined to be strongly stable somewhere in the lowest 1 km, but the height of this strongly stable layer can become elevated, separated from the surface by a well-mixed layer, when turbulence is generated.

The analysis presented here finds that faster wind speeds occur when weaker stability regimes are present (Fig. 5a-e). We suggest that these stronger winds enhance mechanical generation of turbulence which allows for the weaker stability regimes to develop. Largely, it was observed that SS, MS, and the VSM regimes occurred when there were below average 2 m wind speeds, and the WS and NN regimes occurred when there were above average 2 m wind speeds. This difference was even more pronounced in winter, suggesting that near-surface wind speed, and subsequent mechanical mixing, is a more important driver of ABL turbulence in winter than in the other seasons. For a given near-surface stability regime, 2 m wind speed increases as stability aloft increases, which suggests that when stability aloft is stronger, more mechanically generated turbulence, and thus faster near-surface wind speeds, are necessary to mix out the near-surface layer. Differences in wind speed in the central Arctic may be explained by the synoptic setting, inferred with the 2 m pressure (Fig. 5f-j), where lower pressure (suggestive of a stormy setting with faster wind speeds) occurred in conjunction with weaker stability, with the most drastic jump down in pressure values being between the VSM (pressure values largely above average) and WS (pressure values largely below average) regimes, which again was most pronounced in winter.

This study also finds a significant difference in radiation budget terms for different stability regimes (Fig. 6, S3, and S4). We suggest that enhanced radiation (i.e., higher amounts of downwelling longwave and/or shortwave radiation, and thus higher net radiation values) at the surface contributes to thermodynamic generation of turbulence which allows for the weaker stability regimes to form. Over the course of the year, the radiative balance at the *Polarstern* was negative, though net radiation was greatest for the VSM and NN regimes (which had net radiation largely above average), whereas SS, MS, and the WS regimes had net radiation largely below average. The VSM and NN regimes were also observed when downwelling longwave and shortwave radiation were above average, where the stronger relationship was between downwelling longwave radiation and stability. For weaker stability aloft, larger radiative fluxes were observed, suggesting that enhanced radiation weakens stability above the ABL just as it weakens near-surface stability. Thus, there is a relationship between stability (both within the ABL and aloft) and net radiation at the surface, which is dominated by the downwelling longwave component, where the VSM and NN regimes were observed when radiation values were higher. Variations in the surface radiation budget can be partly explained by cloud cover, where greater cloud frequency contributes to higher downwelling longwave radiation values (Fig. 7a-e). As this study suggests that enhanced radiation drives turbulent mixing, then increased cloud cover likely weakens ABL stability through enhanced turbulence production at the surface, as well as due to mixing within and below the cloud driven by cloud top cooling. While this study provides a high-level perspective on the interaction between clouds and stability, further research is needed to fully understand the complexities of the relationship. For example, future work could repeat the current study for cloudy versus clear sky conditions, examine the effects of multiple cloud layers, or analyze potential temperature gradients at cloud base height and within a cloud layer as a function of stability or surface net longwave radiation.

When considering both mechanical and radiative influences on stability, it was discovered that, from an annual perspective, SS and MS regimes largely occur in low wind, low radiation (i.e., net and downwelling radiation values are low) environments, the VSM regimes occur in low wind, high radiation (i.e., net and downwelling radiation values

 are high) environments, the WS regimes occur in high wind, moderate radiation environments, and the NN regimes occur in high wind, high radiation environments. Stability aloft increases with increasing wind speeds and decreasing radiation. An exception to the above statement is that, in summer, strong stability was also observed in high downwelling and net radiation conditions, and this strong stability is likely due to advective processes (Tjernström 2005), which manifests in warm moist air advection from over the relatively warmer open ocean (of which there is more in summer) to over the relatively colder sea ice surface, which decouples the colder near-surface atmosphere from the advected layer, resulting in a shallow surface-based inversion and stable ABL. A common signature of this process is fog. This theory is supported by higher fog frequency for stronger stability regimes (Fig. 7f) and greater atmospheric moisture associated with stronger stability (Fig. 8) in summer.

While we discuss the results of this analysis with the assumption that stability occurs as a response to wind and radiation features, we recognize the possibility that wind and radiation features can also occur as a response to stability, and further work is needed to fully understand the complex relationships between stability and the turbulent processes addressed in this paper. One limitation of this study is that stability regimes are based on radiosonde profiles starting at 35 m, since measurements below this are often unreliable, so differences in stability below this height are neglected (and potentially important). A complementary paper (Jozef et al., 2023b) addresses the annual statistics of many of the thermodynamic and kinematic features noted in this study (such as characteristics and frequencies of ABL, low-level jet, temperature inversions, and moisture features), depending on stability regime, to provide an annual cycle of the central Arctic ABL, and thus such results are not addressed in this work. Future work will be conducted to determine how well the observed results are represented by weather and climate models. Thus, we hope that these findings serve to help inform the improvement of parameterizations of the central Arctic in weather and climate models.

**Data availability**

The level 2 radiosonde data used in this study are available at the PANGAEA Data Publisher at https://doi.org/10.1594/PANGAEA.928656 (Maturilli et al., 2021). Meteorological tower data are available at the National Science Foundation Arctic Data Center at https://doi.org/10.18739/A2PV6B83F (Cox et al., 2023a) as described in Cox et al. (2023b). Ceilometer and microwave radiometer data are available at the Department of Energy Atmospheric Radiation Measurement Data Center at http://dx.doi.org/10.5439/1181954 (ARM user facility, 2019a) and http://dx.doi.org/10.5439/1027369 (ARM user facility, 2019b) respectively, as described in Shupe et al. (2021).

**Author contributions**

SD provided the radiosonde data; CC provided the meteorological tower and radiation data; GdB and JC acquired funding for analysis; GJ, JC, MD and GdB conceptualized the analysis presented in this paper; GJ analyzed the data; GJ wrote the manuscript; JC, MD, GdB, SD and CC reviewed and edited the manuscript.

**Competing interests**

The authors declare that they have no conflict of interest.

**Acknowledgments**

Data used in this paper were produced as part of RV *Polarstern* cruise AWI_PS122_00 and of the international Multidisciplinary drifting Observatory for the Study of the Arctic Climate (MOSAiC) with the tag MOSAiC20192020. We thank all those who contributed to MOSAiC and made this endeavor possible (Nixdorf et al., 2021). Radiosonde data were obtained through a partnership between the leading Alfred Wegener Institute (AWI), the Atmospheric Radiation Measurement (ARM) User Facility, a US Department of Energy (DOE) facility managed by the Biological

and Environmental Research Program, and the German Weather Service (DWD). Meteorological tower data were obtained by the National Oceanographic and Atmospheric Administration (NOAA). Ceilometer and microwave radiometer data were obtained by the AWI and DOE-ARM User Facility. Radiation data were obtained by the DOE-ARM User Facility. We appreciate comments provided by an anonymous internal reviewer at NOAA.

**Financial support**

Funding support for this analysis was provided by the National Science Foundation (award OPP 1805569, de Boer, PI) and the National Aeronautics and Space Administration (award 80NSSC19M0194). The meteorological tower and radiation observations were supported by the National Science Foundation OPP-1724551, by NOAA's Physical Sciences Laboratory (PSL) (NOAA Cooperative Agreement NA22OAR4320151) and by NOAA's Global Ocean Monitoring and Observing Program (GOMO)/Arctic Research Program (ARP)

(FundRef https://doi.org/10.13039/100018302). Additional funding and support were provided by the Department of Atmospheric and Oceanic Sciences at the University of Colorado Boulder, the Cooperative Institute for Research in Environmental Sciences, the National Oceanic and Atmospheric Administration Physical Sciences Laboratory, and the Alfred Wegener Institute.

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
