# Peer review of "Thermodynamic and Kinematic Drivers of Atmospheric Boundary Layer Stability in the Central Arctic during MOSAiC"

_EGUsphere, 2023_

## Referee Comment (RC1)

**Reviewer comments for: Thermodynamic and Kinematic Drivers of Atmospheric Boundary Layer Stability in the Central Arctic during MOSAiC**

Gina C. Jozef et al.
*Atmospheric Chemistry and Physics*

**Recommendation: Minor Revisions**

**General Comments**

This paper presents a comprehensive overview of the distributions of atmospheric boundary layer (ABL) stability during the MOSAiC field campaign by classifying observed radiosonde vertical profiles into 1 of 12 stability regimes. The authors also thoroughly detail physical (thermodynamic and kinematic) explanations for these observed distributions, both in a bulk annual sense and by accunting for seasonal variations in the Arctic. Overall, this paper is organized very well, the discussions are scientifically sound, and the writing style is clear and concise. I especially appreciate the situational awareness that was demonstrated when it came to instrument placement during fog events and how a synthesis of observations can show a more complete picture. These results are certainly pertinent to future studies of the lower atmosphere during the MOSAiC campaign, making this paper a suitable fit for the journal ACP. I am pleased to recommend this paper for publication after the authors address a handful of minor comments that are outlined below.

**Minor and Technical Comments**

1. Line 53: The phrase "negative longwave balance" is somewhat contradictory, perhaps change to read "net negative longwave radiation at the surface."

2. Line 123: Perhaps I missed it, but please define the acronym "ARM" somewhere in the text before using it here.

3. Line 184: The sentence seems to awkwardly break with the phrase "..., within the ABL, ..." Since you already describe the criteria for $d\theta_v/dz$ to be near the surface, I think the qualifier "within the ABL" can be omitted here.

4. Section 2.3: I think this paper would strongly benefit from the inclusion of example profiles from some or all of the stability regimes outlined in this section and in Table

2. These could be either synthetic data with linear profiles in each altitude range considered, or they could also be an example profile from real data that exemplify the criteria for each regime. Because there are not too many figures already, please consider including an additional figure to go along with this section, as I think this will help readers more firmly grasp the physical arguments discussed throughout. The example profiles can also be color-coded to match the same color scheme used throughout this paper for consistency.

5. Section 2.3: In general, I think it would be useful to contextualize the stability regime criteria with others in the literature based on parameters such as the Richardson number or a layer-specific lapse rate (see, for example, Sorbjan, 2010; Sorbjan and Grachev, 2010; Pithan et al., 2014). Additionally, I think it would be interesting to consider the joint distributions of surface net radiation and bulk ABL lapse rates for quasi-direct comparison with those by Pithan et al. (2014) using data from the SHEBA campaign.

6. Table 2: In the first column header, it seems $D\theta_v/dz$ should rather read as $d\theta_v/dz$ for notation consistency.

7. Table 2 and throughout: I appreciate the consistent use of the color scheme throughout the paper for classifying each stability regime. However, please consider using a colorblind-friendly alternative to the red/green/orange base palette utilized throughout the paper.

8. Lines 316–338: The logic in using pressure as a proxy for synoptic setup seems reasonable to me, but would pressure tendency $\partial p/\partial t$ be a more useful proxy for the onset of storm systems in this case? The sign may also indicate whether a storm is approaching or receding, so if this is too granular for the purposes of this study, maybe even just the magnitude of the pressure tendency could be useful. Please discuss.

9. Line 364: When stating that the "...interquartile ranges of net radiation for ... regimes exceeded zero," to my understanding this means the $75^{\text{th}}$ percentiles exceed zero. Am I correct in this reasoning? Please clarify.

10. Line 385: Please remove the "a" so the first full sentence reads "This all suggests that longwave radiation is more coupled to ABL stability..."

11. Lines 552–556: As discussed previously in the paper, the causal relationship between surface net longwave radiation and stability within and above the ABL is difficult to determine in a bulk statistical sense such as that presented here. With the given dataset, is it possible to determine a distribution of, e.g., $\partial\theta_v/\partial z$ at cloud base height as a function of stability class or surface net longwave radiation? This may provide additional context in the role that clouds play in destabilizing the lower atmosphere. This analysis is not critical to include, but at the very least I think an additional discussion similar to that provided at lines 281–286 is warranted here in the summary and conclusions section.

**References**

Pithan, F., B. Medeiros, and T. Mauritsen, 2014: Mixed-phase clouds cause climate model biases in Arctic wintertime temperature inversions. *Climate Dynamics*, **43 (1)**, 289–303, https://doi.org/10.1007/s00382-013-1964-9.

Sorbjan, Z., 2010: Gradient-based scales and similarity laws in the stable boundary layer. *Quarterly Journal of the Royal Meteorological Society*, **136 (650)**, 1243–1254, https://doi.org/10.1002/qj.638.

Sorbjan, Z., and A. A. Grachev, 2010: An Evaluation of the Flux–Gradient Relationship in the Stable Boundary Layer. *Boundary-Layer Meteorology*, **135 (3)**, 385–405, https://doi.org/10.1007/s10546-010-9482-3.

---

## Author Comment (AC1)

**Response to Anonymous Referee 1 Comments**

We would like to sincerely thank Anonymous Referee 1 for taking the time to read our manuscript and provide their helpful comments. These comments have helped to significantly improve the manuscript. Each referee comment is given below in ***bold italics*** followed by our response to the comment. The line numbers provided in our responses refer to line numbers in the revised manuscript.

***General Comments***
***This paper presents a comprehensive overview of the distributions of atmospheric boundary layer (ABL) stability during the MOSAiC field campaign by classifying observed radiosonde vertical profiles into 1 of 12 stability regimes. The authors also thoroughly detail physical (thermodynamic and kinematic) explanations for these observed distributions, both in a bulk annual sense and by accunting for seasonal variations in the Arctic. Overall, this paper is organized very well, the discussions are scientifically sound, and the writing style is clear and concise. I especially appreciate the situational awareness that was demonstrated when it came to instrument placement during fog events and how a synthesis of observations can show a more complete picture. These results are certainly pertinent to future studies of the lower atmosphere during the MOSAiC campaign, making this paper a suitable fit for the journal ACP. I am pleased to recommend this paper for publication after the authors address a handful of minor comments that are outlined below.***

Thank you for your positive review of our paper. Below we address each of your comments, and explain how and where changes have been made to the manuscript.

***Minor and Technical Comments***
***1. Line 53: The phrase "negative longwave balance" is somewhat contradictory, perhaps change to read "net negative longwave radiation at the surface."***

This change has been made, and the sentence now reads:

"… there is negative net longwave radiation at the surface…" (line 54)

***2. Line 123: Perhaps I missed it, but please define the acronym "ARM" somewhere in the text before using it here.***

We have added a sentence to introduce ARM (Atmospheric Radiation Measurement mobile facility) before discussing ARM instrumentation and products:

"Several additional measurements come from instrumentation deployed as part of the Atmospheric Radiation Measurement (ARM) mobile facility (Shupe et al., 2021)." (line 122).

***3. Line 184: The sentence seems to awkwardly break with the phrase "..., within the ABL, ..." Since you already describe the criteria for dθv/dz to be near the surface, I think the qualifier "within the ABL" can be omitted here.***

The authors agree that this sentence was phrased poorly. We have restructured the sentence to now read:

"Since the stability criteria in part depend on stability within the ABL and some observations have an ABL height as low as 50 m, we first include a measurement of $d\theta_v/dz$ at 42.5 m (this determines the near-surface stability), calculated across a 15 m interval between 35 m (lowest point of the profile) and 50 m." (line 207)

***4. Section 2.3: I think this paper would strongly benefit from the inclusion of example profiles from some or all of the stability regimes outlined in this section and in Table 2. These could be either synthetic data with linear profiles in each altitude range considered, or they could also be an example profile from real data that exemplify the criteria for each regime. Because there are not too many figures already, please consider including an additional figure to go along with this section, as I think this will help readers more firmly grasp the physical arguments discussed throughout. The example profiles can also be color-coded to match the same color scheme used throughout this paper for consistency.***

Thank you for this very nice suggestion. The authors now include a figure in Sect. 2.3 (Fig. 2 of the revised manuscript) which shows an example profile for each stability regime from the radiosonde data, aside from NN, as there are no purely NN cases in the observations. The example profiles show both the $d\theta_v/dz$ and $\theta_v$ anomaly profiles for each example (line 264). We also have added some text introducing the new figure:

"All of the resulting options for stability regime are listed in Table 2, and an example case for each regime (except NN) is shown in Fig. 2. The color-coding in Table 2 will be used to discern each regime henceforth. While we list NN as a stability regime option, a purely NN case without enhanced stability aloft was never observed in a MOSAiC radiosonde profile, and as such no NN example is given in Fig. 2." (line 251).

***5. Section 2.3: In general, I think it would be useful to contextualize the stability regime criteria with others in the literature based on parameters such as the Richardson number or a layer-specific lapse rate (see, for example, Sorbjan, 2010; Sorbjan and Grachev, 2010; Pithan et al., 2014). Additionally, I think it would be interesting to consider the joint distributions of surface net radiation and bulk ABL lapse rates for quasi-direct comparison with those by Pithan et al. (2014) using data from the SHEBA campaign.***

We have added some text in the beginning of Sect. 2.3 to explain how the definition of stability regime in the current manuscript expands upon that used in prior literature:

"By defining twelve distinct stability regimes, we expand upon the traditional categorization of stability into one of three categories: stable, neutral, and unstable (Stull, 1988; Liu and Liang, 2010). While some prior studies have separated the stable regime into a few subcategories for the Arctic (weakly stable, very stable, and extremely stable; Sorbjan, 2010; Sorbjan and Grachev, 2010), our analysis expands upon this through the inclusion of additional subcategories for stability above the ABL." (line 192)

We also added some text toward the end of Sect. 2.3 which mentions the methods of stability regime identification used in previous literature (including Sorbjan (2010;), Sorbjan and Grachev (2010), and Pithan et al. (2014)), to contrast with the methods of the current paper, and in the end explain why the current methods were chosen:

"While other studies define stability in the Arctic based on $Ri_b$ and local Obukhov length (Sorbjan, 2010; Sorbjan and Grachev, 2010), or based on temperature lapse rate (Pithan et al., 2014), we found the above methods for defining stability regime based on $d\theta_v/dz$ and ABL height to yield reliable results while providing the best potential for repeatability in future work (e.g., Dice et al., submitted), as the methods rely only on standard radiosonde observations (and do not require additional measurements). This also allows us to apply the same methods to both the near-surface and aloft stabilities. Additionally, as the focus of this study is to analyze the relationships between turbulent forcing mechanisms and stability, metrics for stability regime identification that include these forcing mechanisms in their definition (e.g., Obukhov length and $Ri_b$ include wind speed in their calculations) were avoided. Comparison of the stability regimes determined using the methods described in this section to bulk friction velocity from the met tower (Jozef et al., 2023b) shows that the current methods discern meaningful differences in turbulence between the various stability regimes." (line 240)

Lastly, we have created a figure comparable to the bivariate pdf comparing low level atmospheric lapse rate to surface net LW radiation shown in Fig. 10 in Pithan et al. (2014). See the figure below, which is a binned scatter plot of $d\theta_v/dz$ over the depth of the ABL vs. surface net LW during MOSAiC. A similar bimodal distribution is seen in the MOSAiC data as in Pithan et al. (2014), in which there is a cluster of cases with surface net LW less than -25 W/m2 (the clear sky state) and surface net LW greater than -25 W/m2 (the cloudy state), where stability decreases (indicated by a decreasing $d\theta_v/dz$) with greater longwave cooling (lower surface net LW values). This is the same result as was presented in Pithan et al. (2014). It was a nice suggestion to add this analysis. Thus, we have added some discussion in the text to indicate that the MOSAiC data shows the same results as Fig. 10 in Pithan et al. (2014).

"Comparison of surface net longwave radiation to ABL stability reveals that there is a bimodal distribution with weaker stability more often occurring in the cloudy sky mode (surface let longwave greater than -25 W m2) and stronger stability more often occurring in the clear sky mode (surface let longwave less than -25 W m2). Further, within the clear sky mode, stronger stability corresponds to weaker longwave cooling. These results agree with Pithan et al. (2014) which revealed these conclusions using data from the SHEBA project." (line 437)

[Figure]

We additionally include another statement where the results of the current paper agree with the results from Pithan et al. (2014):

"This again agrees with the results of Pithan et al. (2014) which showed that stability is stronger in the Arctic clear sky state." (line 510).

**6. Table 2: In the first column header, it seems Dθv/dz should rather read as dθv/dz for notation consistency.**

Thank you for pointing this out. The authors had missed this auto capitalization of the table heading and only noticed after the manuscript had been submitted. It has now been corrected to $d\theta_v/dz$ (line 256).

**7. Table 2 and throughout: I appreciate the consistent use of the color scheme throughout the paper for classifying each stability regime. However, please consider using a colorblind-friendly alternative to the red/green/orange base palette utilized throughout the paper.**

We have changed the color scheme to be more color-blind friendly. While it is impossible to find 12 colors that look distinct for all possible colorblindness options, we do at least now separate out the colors that look similar to colorblind people so that they don't appear next to each other. Additionally, in order to accommodate for colorblindness, we always include a legend that lists the stability regimes in the same order as they appear on the figure, so one could always determine which stability regime they are looking at based on its location among all stability regime options.

**8. Lines 316–338: The logic in using pressure as a proxy for synoptic setup seems reasonable to me, but would pressure tendency ∂p/∂t be a more useful proxy for the onset of storm systems in this case? The sign may also indicate whether a storm is approaching or receding, so if this is too granular for the purposes of this study, maybe even just the magnitude of the pressure tendency could be useful. Please discuss.**

We have added a panel to the figure showing 2 m wind speed and pressure depending on stability regime and season (Figure 5 in the updated manuscript, line 401) which shows the absolute pressure tendency (dp/dt) calculated as the change in hPa over the 3 hours preceding each observation, as well as corresponding significance testing in the supplementary figures (Supplementary Figure S2 in the updated manuscript). We found similar trends in dp/dt as were found in the 2 m pressure, in that dp/dt is greater for the weaker stability regimes, further supporting the theory that synoptic scale storm systems contribute to the higher wind speed events that correspond to the weaker stability regimes. We have added some discussion on the dp/dt results throughout Sect. 3.3:

"Figure 5f-j shows the range of 2 m pressure and Fig. 5k-o shows the range of absolute 2 m pressure tendency (dp/dt) corresponding to each radiosonde launch for each stability regime and season (refer to Supplementary Fig.

S2 for corresponding significance testing), where the annual mean of 2 m pressure and dp/dt throughout MOSAiC were 1010.8 hPa and 0.77 hPa (3 hr)$^{-1}$ respectively. Annually, the pressure results mimic what was seen with 2 m wind speed, in that lower pressure and greater dp/dt (suggestive of a stormy setting with faster wind speeds) is correlated with weaker stability, with the most drastic reduction in pressure and increase in dp/dt values being between SS, MS and the VSM regimes (pressure largely above average and dp/dt largely below average) and the WS and NN regimes (pressure largely below average and dp/dt largely above average; difference in means of 6.6 hPa and 0.31 hPa (3 hr)$^{-1}$ respectively). This is supported by Supplementary Fig. S2a which shows a high level of significance when comparing 2 m pressure and dp/dt between different stability regimes." (line 375).

"Differences in 2 m dp/dt between stability regimes in winter are not as great as annually or in fall or spring (Fig. 5m and S2c) suggesting more slowly evolving low and high pressure systems in winter than in other seasons. The smallest differences in 2 m pressure and dp/dt between stability regimes occurred in summer (Fig. 5j and S2e), again echoing the results from the 2 m wind speed, and further supporting the statement that the presence of storms, and resulting wind shear, are not the most important drivers of ABL stability in summer." (line 394).

The authors did not want to replace the 2 m pressure results with the dp/dt results, as dp/dt only tells us about the onset or diminishment of a storm, but can miss periods during a storm when the low pressure persists. Thus, by including both 2 m pressure and dp/dt, we get the full story, and our argument is strengthened.

***9. Line 364: When stating that the "...interquartile ranges of net radiation for ... regimes exceeded zero," to my understanding this means the 75th percentiles exceed zero. Am I correct in this reasoning? Please clarify.***

You are correct in this reasoning. To clarify this, we now state:
"… the 75$^{th}$ percentiles (upper limit of the interquartile range) of net radiation… exceed zero…" (line 427)

***10. Line 385: Please remove the "a" so the first full sentence reads "This all suggests that longwave radiation is more coupled to ABL stability..."***

Thank you for pointing out this error. The "a" has been removed so the sentence now reads:

"This all suggests that longwave radiation is more coupled to ABL stability throughout the span of the year than shortwave radiation." (line 454)

***11. Lines 552–556: As discussed previously in the paper, the causal relationship between surface net longwave radiation and stability within and above the ABL is difficult to determine in a bulk statistical sense such as that presented here. With the given dataset, is it possible to determine a distribution of, e.g., ∂θv/∂z at cloud base height as a function of stability class or surface net longwave radiation? This may provide additional context in the role that clouds play in destabilizing the lower atmosphere. This analysis is not critical to include, but at the very least I think an additional discussion similar to that provided at lines 281–286 is warranted here in the summary and conclusions section.***

We have added some text in the summary and conclusion section stating that:

"While we discuss the results of this analysis with the assumption that stability occurs as a response to wind and radiation features, we recognize the possibility that wind and radiation features can also occur as a response to stability, and further work is needed to fully understand the complex relationships between stability and the turbulent processes addressed in this paper." (line 649)

Analysis of dθ$_v$/dz at cloud base height as a function of stability class or surface net longwave radiation is an interesting idea, but (as you also mentioned) the authors believe this to be outside of the scope of the current paper, as one could spend a whole paper exploring in more detail the interactions between clouds, radiation, and stability, whereas this paper is intended to give a more broad perspective on the subject. It would be useful for a future study to isolate only cloudy cases and perform the analysis you have suggested. We have added some text to the summary and conclusions section stating that the results of the current study only scratch the surface of the relationship between clouds and stability, and have suggested some areas for future research:

"While this study provides a high-level perspective on the interaction between clouds and stability, further research is needed to fully understand the complexities of the relationship. For example, future work could repeat the current study for cloudy versus clear sky conditions, examine the effects of multiple cloud layers, or analyze potential temperature gradients at cloud base height and within a cloud layer as a function of stability or surface net longwave radiation." (line 632).

*References*
- *Pithan, F., B. Medeiros, and T. Mauritsen, 2014: Mixed-phase clouds cause climate model biases in Arctic wintertime temperature inversions. Climate Dynamics, 43 (1), 289–303, https://doi.org/10.1007/s00382-013-1964-9.*
- *Sorbjan, Z., 2010: Gradient-based scales and similarity laws in the stable boundary layer. Quarterly Journal of the Royal Meteorological Society, 136 (650), 1243–1254, https://doi.org/10.1002/qj.638.*
- *Sorbjan, Z., and A. A. Grachev, 2010: An Evaluation of the Flux–Gradient Relationship in the Stable Boundary Layer. Boundary-Layer Meteorology, 135 (3), 385–405, https://doi.org/10.1007/s10546-010-9482-3.*

---

## Author Comment (AC2)

**Response to Anonymous Referee 2 Comments**

We would like to sincerely thank Anonymous Referee 2 for taking the time to read our manuscript and provide their helpful comments. These comments have helped to significantly improve the manuscript. Each referee comment is given below in ***bold italics*** followed by our response to the comment. The line numbers provided in our responses refer to line numbers in the revised manuscript.

*General*

***This paper investigates atmospheric stability conditions over the Arctic Ocean using sounding data obtained during the year-long MOSAiC drift of the research vessel Polarstern. Based on a classification of stability they elaborate thermodynamic and kinematic drivers of the ABL structure.***

***The paper is very well organized, in most parts well written and presents interesting findings, which are new and will stimulate probably further research. Nevertheless, I suggest some major revisions before its publication.***

Thank you for your review of our paper. Below we address each of your comments, and explain how and where changes have been made to the manuscript.

*Major Revisions*

***1) My most important point concerns the definition of stability. The paper lives from the definition of classes, whose motivation is described in detail in a paper that is not yet finally accepted for publication. For this reason, I find it necessary to obtain more details here. E.g. the question arises if the boundaries of the classes have physical reasons. The definition of weakly stable, strongly stable and so on has long tradition (e.g. Mahrt, 1998), so it needs motivation when these terms are used in a new sense and with new thresholds.***

The paper that contains more details on the motivations for the specific methods and thresholds used for stability classification is now in preprint (Jozef et al., 2023a, https://doi.org/10.5194/essd-2023-141), so it can be referred to while reading the current manuscript. We have however added the details of the motivations for the stability regime classification and thresholds to the current paper:

[revised manuscript text omitted]

We have also added some text at the end of Sect 2.3 that describes why the current methods were used rather than those using Obukhov length (as in Mahrt, 1998):

"While other studies define stability in the Arctic based on $Ri_b$ and local Obukhov length (Sorbjan, 2010; Sorbjan and Grachev, 2010), or based on temperature lapse rate (Pithan et al., 2014), we found the above methods for defining stability regime based on $d\theta_v/dz$ and ABL height to yield reliable results while providing the best potential for repeatability in future work (e.g., Dice et al., submitted), as the methods rely only on standard radiosonde observations (and do not require additional measurements). This also allows us to apply the same methods to both the near-surface and aloft stabilities. Additionally, as the focus of this study is to analyze the relationships between turbulent forcing mechanisms and stability, metrics for stability regime identification that include these forcing mechanisms in their definition (e.g., Obukhov length and $Ri_b$ include wind speed in their calculations) were avoided. Comparison of the stability regimes determined using the methods described in this section to bulk friction velocity from the met tower (Jozef et al., 2023b) shows that the current methods discern meaningful differences in turbulence between the various stability regimes." (line 240-250)

***2) In this connection it is important that a physical classification of the surface layer is usually done in terms of z/L where z is height and L is the Obukhov length. L does not seem to be available here (or is not yet available?). The authors investigate just the thermal stability, while dynamical stability is more important for modeling since flux parameterizations depend on it. This investigation could be done in terms of the dependence of the (bulk) Richardson number, which would include the effect of wind speed. Is there a reason why this was avoided? I am not against the consideration of thermal stability but it might be worth to consider the Richardson number dependent stability in addition. At least, a discussion would help in Section 3.3 when the wind speed dependence is studied.***

The stability regime classifications in this study build upon methods of previous studies which discern stability using potential temperature gradient (Stull, 1988; Liu and Liang, 2010; Dice and Cassano, 2022, Jozef et al., 2022). This is a similarly well-recognized and documented method for stability regime identification, just as is z/L.

Potential temperature gradient was chosen for this study over z/L because z/L is only really applicable for the surface layer, and we are similarly interested in stability aloft, while applying the same methods/thresholds to the near-surface and aloft layers. Also, L would depend on tower measurements, and our goal is to use methods which only rely on the radiosonde measurements to allow application to a wide range of sites that may lack in-situ

turbulence measurements. $Ri_b$ was also not the chosen metric because $Ri_b$ doesn't well discern differences between the strength of stability in the MOSAiC Arctic data. We did look at $Ri_b$ within the ABL as a function of stability, and found little difference across the different regimes, as $Ri_b$ is always below 0.5 within the ABL (as a function of our ABL height identification methods) and above 0.5 above the ABL, with little consistency in differences in $Ri_b$ for the different stability options.

Additionally, we wanted to compare independent variables in this study, and as z/L and $Ri_b$ directly include wind speed in their definition, it would be difficult to disentangle stability defined by z/L or $Ri_b$ and wind speed as a forcing mechanism. Thus, the results we show in Sect. 3.3 that wind speed differs discernably between stability regimes (defined independently from wind speed) make a reliable and strong case that stability and wind speed are indeed correlated. We have added some text at the end of Sect 2.2 which mentions some methods of stability classification used by previous studies, and explains why the current methods were chosen instead:

"While other studies define stability in the Arctic based on $Ri_b$ and local Obukhov length (Sorbjan, 2010; Sorbjan and Grachev, 2010), or based on temperature lapse rate (Pithan et al., 2014), we found the above methods for defining stability regime based on $d\theta_v/dz$ and ABL height to yield reliable results while providing the best potential for repeatability in future work (e.g., Dice et al., submitted), as the methods rely only on standard radiosonde observations (and do not require additional measurements). This also allows us to apply the same methods to both the near-surface and aloft stabilities. Additionally, as the focus of this study is to analyze the relationships between turbulent forcing mechanisms and stability, metrics for stability regime identification that include these forcing mechanisms in their definition (e.g., Obukhov length and $Ri_b$ include wind speed in their calculations) were avoided. Comparison of the stability regimes determined using the methods described in this section to bulk friction velocity from the met tower (Jozef et al., 2023b) shows that the current methods discern meaningful differences in turbulence between the various stability regimes." (line 240-250)

We have also added some text in Sect 3.3 which states that it is well-recognized that stability is dependent on wind speed (reflected through the definition of $Ri_b$ which is often used for stability classification), which is even better supported by the results of our study, as our stability classification is not directly dependent on wind speed values:

"… as 2 m wind speed increases, near-surface stability decreases, indicating that wind speed is correlated to ABL stability. This agrees with the well-documented notion that stability is dependent on wind speed (Brooks et al., 2017; Banta et al., 2003), as is reflected in the definition of $Ri_b$ (Stull, 1988), which is often used as a metric for determining stability. As stability regime classification in the current study is not directly dependent on wind speed, evidence for this relationship is strengthened." (line 348).

*3) I think the description of the Table 2 needs improvement. I have a list of questions here:*

*Line 190: Why is the near-surface stability representative of the stability within the entire ABL? Decoupling might occur etc….*

The "near-surface stability" that we refer to is that at an altitude of 42.5 m. Traditionally, the ABL is defined by stability near the surface, which is determined in this study by calculating $d\theta_v/dz$ over the lowest layer of the radiosonde observations. Thus, the stability at 42.5 m defines of the ABL stability. We have added clarification that "near-surface stability" for our purposes refers to stability of the ABL as dictated by the $d\theta_v/dz$ value at 42.5 m (which is not necessarily the same stability as throughout the entire ABL):

"Twelve stability regimes have been defined based on stability within the ABL (hereafter referred to as "near-surface" stability)…" (line 190)

"The first step for stability regime identification is to classify the near-surface stability using the $d\theta_v/dz$ value at 42.5 m. As the ABL at any given location is defined by the stability near the surface (Stull, 1988), this $d\theta_v/dz$ value at 42.5 m reasonably indicates the ABL stability." (line 212)

*In line 185 it is said that the lowest value is between 15 and 35 m, but in line 190 the near-surface stability is based on the 42.5 m level. Both together is puzzling.*

Apologies for this confusion. The lowest value is calculated between 35 m and 50 m (which is a difference of 15 m), such that the center point of that interval is 42.5 m. We have clarified the text by now saying that:

"… we first include a measurement of $d\theta_v/dz$ at 42.5 m (this determines the near-surface stability), calculated across a 15 m interval between 35 m (lowest point of the profile) and 50 m." (line 208).

*First row of Table2 : Does this refer to ABL heights smaller than 50 m ?*

While the ABL height of SS cases is often 50 m, this is not necessarily true in every case, nor is it relevant for the methods, as we state that ABL height is only considered for the differentiation of a WS or NN case into the VSM category. To clarify, we have added a sentence that states:

"The ABL height is not relevant for the definition of SS and MS, though these regimes usually have an ABL height less than 125 m, and SS cases often have an ABL height as low as 50 m." (line 231)

*I understood that the term 'aloft' refers to the layer above the ABL but below 1km. But is the given stability then an average over the whole layer? I am asking because often there is a capping inversion with strong stability but in upper layers stability is weaker.*

We have clarified the wording to state that stability "aloft" is based on:

"…the strength of the capping $\theta_v$ inversion located between the top of the ABL and 1 km" (line 191).

As such, stability aloft is not an average over the whole layer, but is rather defined by the strongest stability layer located between the ABL and 1 km (i.e., the capping $\theta_v$ inversion).

*Why is unstable stratification not a part of the classification? SHEBA data showed such cases.*

In the revised manuscript, we now describe that the 12 stability regime categories were determined using the range of $\theta_v$ vertical structures revealed by a self-organizing map (SOM) analysis in Jozef et al. (2023), trained using the year of MOSAiC radiosonde data. Of the 30 SOM patterns, not a single one represented an unstable atmosphere, so it was concluded that the frequency of instability above 35 m during MOSAiC was minimal enough to not be relevant to the current study. This is consistent with Persson et al. (2002) which analyzed near surface stability during SHEBA and found that in the case of unstable conditions, the atmosphere was only typically unstable below about 5 m, and then near-neutral or stable above that. Thus, unstable conditions would usually be too low level to be recognized by the methods of the current study. Any cases with instability above 35 m are very rare, and thus are grouped into the near-neutral (NN) category. We have added a sentence which states:

"Near-surface instability is not considered as its own category, as the instances are very few, and any such cases are grouped into the NN category." (line 216)

*Minor Revisions*

*1) During MOSAiC, also turbulence was measured at a mast taller than 2 m, are the data not yet available? These data would help for the stability classification, but also to correct the biases of the soundings near the surface that are discussed in section 2.2 (?)*

The authors did consider use of the 10 m tower and 30 m mast also deployed during MOSAiC for correcting biases in the near surface radiosonde measurements, which would help with stability classification. These measurements

were used to some extent, but were found ultimately to not be useful for other reasons. Below we summarize the points considered:

- 10 m tower (referred to as "met tower" in the manuscript) measurements were used to determine when low level radiosonde measurements were likely incorrect. This allowed us to determine the altitude of the lowest reliable measurement for each radiosonde profile. This was already described in Sect. 2.2 of the original manuscript (line 143-149), and is retained in the revised manuscript (line 147-156).
- We also considered merging the tower and mast measurements with the radiosonde measurements, but found a frequent offset in temperature between the tower/mast and the radiosonde which could be a result of slight changes in the atmospheric properties and surface state between the locations of the tower/mast and radiosondes, differences in instrument accuracy/uncertainties, etc. Merging the two products could create false gradients in the lower atmosphere such that stability could be inaccurately identified.
- Using only the tower/mast measurements would not tell us about stability aloft, which is a goal of the current study. One could argue that we could use the tower/mast turbulence measurements to tell us about the near-surface stability, and the radiosondes to tell us about aloft stability, but the authors wanted to develop methods which would consistently apply the same methods for determining the near surface and aloft stabilities alike.
- One of the goals for our stability classification methods is that they are repeatable at other locations. Not all campaigns can expect to have both a tower/mast and radiosonde that are deployed close together such that the products can be used in conjunction. Therefore, we sought to use stability classification methods that rely solely on the radiosonde observations. Specifically, these stability regime classification methods were developed in conjunction with a similar study being conducted across the Antarctic continent (at both coastal and inland sites) which don't always have tower/mast measurements to use, so that common methodology that works well at both poles was applied.
- In the complementary paper referenced in the current manuscript (Jozef et al., 2023b), we look at bulk friction velocity from the 10 m tower (a direct metric for turbulence) in the context of stability regime, and see discernable differences across the range of stability regimes (where friction velocity increases with decreasing stability). This gives us confidence that our stability regime classifications does well differentiate between stability in a meaningful way.

The authors have added some text throughout the manuscript to make it clear to the audience that the above points have been considered. Specifically, we now explain why the met tower/radiosonde measurements were not simply merged:

"…met tower measurements were not merged to the radiosonde measurements due to frequent temperature offsets which could occur as a result of the two platforms sampling a slightly different airmass, differences in surface state, differences in instrument accuracy/uncertainty, etc…" (line 152)

We also explain why we developed methods which rely solely on radiosonde measurements, the validity of which have been confirmed through bulk friction velocity from the met tower:

"While other studies define stability in the Arctic based on $Ri_b$ and local Obukhov length (Sorbjan, 2010; Sorbjan and Grachev, 2010), or based on temperature lapse rate (Pithan et al., 2014), we found the above methods for defining stability regime based on $d\theta_v/dz$ and ABL height to yield reliable results while providing the best potential for repeatability in future work (e.g., Dice et al., submitted), as the methods rely only on standard radiosonde observations (and do not require additional measurements). This also allows us to apply the same methods to both the near-surface and aloft stabilities." (line 240)

"Comparison of the stability regimes determined using the methods described in this section to bulk friction velocity from the met tower (Jozef et al., 2023b) shows that the current methods discern meaningful differences in turbulence between the various stability regimes." (line 247).

***2) The quality of some figures (4, 5, 7) is limited. Please improve the resolution and/or increase their size.***

We have improved the resolution of these (and all other) figures.

**Line 160) Please write clearly that measurements below 35 m are not used (as in the Summary section).**

The sentence now reads:

"To allow for a consistent bottom height for our analysis, we only consider profiles in which there is a good measurement at 35 m, and do not consider data at altitudes below 35 m." (line 163).

**Line 164: There are different definitions of $Ri_b$ in the literature. So, it should be defined here.**

We have added additional details about the definition of Rib used in the current study:

"$Ri_b$ was calculated using the following equation from Stull (1988):

$$\mathrm{Ri_b}(z) = \frac{\left(\frac{g}{\overline{\theta_v}}\right)\Delta\theta_v\,\Delta z}{\Delta u^2 + \Delta v^2} \tag{1}$$

where $g$ is acceleration due to gravity, $\overline{\theta_v}$ is the mean virtual potential temperature over the altitude range being considered, $z$ is altitude, $u$ is zonal wind speed, $v$ is meridional wind speed, and $\Delta$ represents the difference over the altitude range used to calculate $Ri_b$ throughout the profile. $Ri_b$ profiles were created by calculating $Ri_b$ across 30 m intervals in steps of 5 m (Jozef et al., 2023a)." (line 170)

**Line 225: replace 'trends' by 'tends'**

This change has been made:

"…the central Arctic lower atmosphere tends towards being strongly stable, but sometimes the near-surface atmosphere can become well-mixed due to the generation of turbulence." (line 281).

**Line 293: 'wind speed correlated with stability'. Any other finding would be very strange (see definition of $Ri_b$).**

You are correct that this is an expected result due to the well documented relationship between wind speed and stability. Nonetheless, we think it is important to state that we see this phenomenon in the current observations, before going into further detail on the relationships between our defined stability regimes and observed wind speeds. It should be noted that a similar study in Antarctica (Dice et al, submitted) finds increasing wind speed with increasing stability at some sites, so this expected relationship does not always hold. To your point, we added some remarks about how this is an expected result, in agreement with the well-recognized relationship between stability and wind speed which is reflected in the definition of $Ri_b$:

"This agrees with the well-documented notion that stability is dependent on wind speed (Brooks et al., 2017; Banta et al., 2003), as is reflected in the definition of $Ri_b$ (Stull, 1988), which is often used as a metric for determining stability." (line 349).

**Lines 308-310: Isn't that very similar in spring? Furthermore, my impression is that the wind speed distribution shows only little dependence on the seasons, and perhaps this whole paragraph can be shortened.**

To simplify this paragraph, we have removed mention of how the results in fall and spring differ from the annual results, as you are right, they are both very similar to the annual trend, and not really worth mentioning. We do wish to retain the discussion on how the winter and summer differ from the annual results and from each other, as we

think it is notable that wind speeds for the various stability regimes differ more from each other in winter than they do in summer:

"Seasonally, there was little difference from the annual pattern, however there are some notable discrepancies. In winter (Fig. 5c), there is a larger increase in 2 m wind speed between SS, MS and the VSM regimes and the WS and NN regimes (increase of 3.7 m s$^{-1}$ versus 3.3 m s$^{-1}$ annually), and a greater number of regimes that have significantly different values from each other (Fig. S1c), suggesting that near-surface wind speed is a more important driver of ABL stability in winter than the other seasons. In summer, there is a smaller increase in 2 m wind speed between SS, MS and the VSM regimes and the WS and NN regimes (increase of 2.7 m s$^{-1}$ versus 3.3 m s$^{-1}$ annually; Fig. 5e), but there is still high significance (Fig. S1c) in the difference between the stronger stability regimes (SS, MS and VSM) to the weaker stability regimes (WS and NN). Thus, while in summer wind shear may not be the most important variable differentiating stability, it still plays a significant role." (line 363).

***Lines 359-362: It is stressed that for VSM and NN another factor than radiation is the driving factor. So, do you conclude that for all other regimes radiation is the most important factor and those discussed in the previous sections are not that important? This needs clarification.***

The point we are trying to make is that VSM and NN both occur under high radiation conditions, so radiation is an important factor for both of these regimes, but there is another factor that comes into play to influence whether the stability is VSM or the stability is NN (this factor being wind speed, as shown in Sect. 3.3). That is not to say that radiation is any more or less of an important factor for the other stability regimes, the other regimes simply occur under lower radiation conditions, but radiation is an important factor for all regimes. To clarify, the sentence now reads:

"This suggests that the VSM and NN regimes similarly occur under higher net radiation conditions (i.e., less negative and sometimes positive), but under these high radiation conditions there is an additional factor that dictates whether stability develops into VSM or instead develops into NN." (line 423).

***Line 385: replace 'is a more' coupled by is 'more coupled'***

Thank you for pointing out this error. The "a" has been removed so the sentence now reads as:

"This all suggests that longwave radiation is more coupled to ABL stability throughout the span of the year than shortwave radiation" (line 454).

***Figure 7 and its description: what is mixing ratio at ABL height? There are often large differences between mixing ratio below and above the ABL top. So, is the value at ABL height still in the ABL (below the inversion)? Does a small variation in height change the conclusion?***

For the purposes of this study, the ABL is considered as the base of the elevated inversion layer (see Jozef et al., 2022 for a detailed study on ABL height identification in the central Arctic). This is now clarified in Sect. 2.2 when we first introduce how ABL height is identified in the current study:

"Following the methods of Jozef et al. (2022) and Jozef et al. (2023a), ABL height from each radiosonde profile was determined by identifying the first altitude in which the bulk Richardson number ($Ri_b$) exceeds a critical value of 0.5 and remains above the critical value for at least 20 consecutive meters… This method identifies the ABL height as the bottom of the elevated $\theta_v$ inversion (or the bottom of the layer of enhanced $\theta_v$ inversion strength) for moderately stable to near-neutral conditions, and at the top of the most stable layer for conditions with a strong surface-based $\theta_v$ inversion." (line 168).

We have also added some text when mixing ratio at ABL height is introduced to specify that mixing ratio at ABL height refers to:

"… the mixing ratio just below the elevated $\theta_v$ inversion or layer of enhanced $\theta_v$ inversion strength" (line 544).

Thus, this should be the mixing ratio at the height where the ABL begins to transition to the capping inversion.

We have recreated the panels for mixing ratio at 15 m below the identified ABL height, and 15 m above the identified ABL height, to see if the conclusions change depending on a small variation in height. These results (shown below) look very similar to the results for mixing ratio at ABL height and thus show that a small variation in height does not change the conclusion. Therefore, the authors do not think it necessary to address this question in the manuscript.

[Figure]

***Line 524: Better 'wide range' for clarity.***

This change has been made:

"…suggesting that these regimes may occur under a wide range of forcing mechanisms." (line 595).

***Line 538: replace 'than the' by 'than in the'***

This change has been made:

"… more important driver of ABL turbulence in winter than in the other seasons." (line 609).

***Line 559: Cloud top cooling enhances turbulence first of all in the center of the cloud and not only below cloud base.***

We have adjusted the sentence to reflect that cloud top cooling also enhances turbulence within the cloud as well as below cloud base. It now reads:

"… as well as due to mixing within and below the cloud driven by cloud top cooling." (line 631)

***Lines 555-564: There is much complexity concerning the cloud-generated mixing, which is only partly described here. Especially in case of multiple cloud layers, decoupling of the lowest ABL may occur. But decoupling of the surface layer can exist also in case of one StCu layer. This might have been captured by the stability classes***

*accounting for the different stability in the surface layer and aloft, so this could be stressed here. Overall, I think, however, that a detailed description of the impact of clouds on stability needs further research and the information given in this section should be seen as one step towards more understanding. E.g., the given stability classification does not distinguish cloud free air and cloudy conditions. Note that, for details, it is the equivalent potential temperature rather than the virtual potential temperature, which would have to be considered for this goal in addition, which is, however, probably beyond the scope of this paper. Some sentences like this could be included.*

We have added some text to the summary and conclusions section stating that the results of the current study only scratch the surface of the relationship between clouds and stability, and have suggested some areas for future research:

"While this study provides a high-level perspective on the interaction between clouds and stability, further research is needed to fully understand the complexities of the relationship. For example, future work could repeat the current study for cloudy versus clear sky conditions, examine the effects of multiple cloud layers, or analyze potential temperature gradients at cloud base height and within a cloud layer as a function of stability or surface net longwave radiation." (line 632).

*Lines 561, 566: 'enhanced radiation'. In the summary it needs clarification, is it longwave, shortwave, net radiation? What is a net radiation regime, what is a low, high radiation environment?*

We now clarify that enhanced radiation refers to:

"…higher amounts of downwelling longwave and/or shortwave radiation, and thus higher net radiation values" (line 618).

We also clarify what is meant by high and low radiation environments when explaining the certain environments which lead to each of the stability regimes:

"… low radiation (i.e., net and downwelling radiation values are low) environments, … high radiation (i.e., net and downwelling radiation values are high) environments" (line 638-640).

Lastly, to avoid confusion in using the word "regimes" when referring to the radiation conditions, we have changed the sentence to read:

"…in summer, strong stability was also observed in high downwelling and net radiation conditions" (line 643).

*References: there are several papers cited, which are submitted or in prep. I do not know if this is against ACP rules. The paper Chechin et al. (2022) was now published in ACP (2023).*

As I understand it, allowable citations in ACP include those in preprint. While some citations referenced in this paper were only in prep or submitted (not yet in preprint) at the time of original submission, all papers originally cited as submitted are now in preprint or published, and the references have been updated accordingly. The Dice et al. (in prep) reference in the original manuscript has now been submitted, and we hope it to be in preprint by the time of publication of the current manuscript. Any papers which are not at least in preprint at the time of acceptance of the current manuscript will be removed from the paper. The Chechin reference has been updated.

(Full references for all other sources mentioned in the author comments can be found in the revised manuscript).